# Northeast Pacific warm blobs sustained via extratropical atmospheric teleconnections

Jian Shi [1,2,3], Hao Huang[1,2], Alexey V. Fedorov [4,5], Neil J. Holbrook [6,7], Yu Zhang [1,3] ✉, Ruiqiang Ding [8], Yongyue Luo [1,2], Shengpeng Wang[3], Jiajie Chen [1,2], Xi Hu[1,2], Qinyu Liu[1], Fei Huang [1,2,3] ✉ & Xiaopei Lin [1,2,3]

Large-scale marine heatwaves in the Northeast Pacific (NEP), identified here and previously as 'warm blobs', have devastating impacts on regional ecosystems. An anomalous atmospheric ridge over the NEP is known to be crucial for maintaining these warm blobs, also causing abnormally cold temperatures over North America during the cold season. Previous studies linked this ridge to teleconnections from tropical sea surface temperature anomalies. However, it was unclear whether teleconnections from the extratropics could also contribute to the ridge. Here we show that planetary wave trains, triggered by increased rainfall and latent heat release over the Mediterranean Sea accompanied by decreased rainfall over the North Atlantic, can transport wave energy to the NEP, guided by the westerly jet, and induce a quasi-barotropic ridge there. Our findings provide insights into extratropical teleconnections sustaining the NEP ridge, offering a source of potential predictability for the warm blobs and temperature fluctuations over North America.

Large-scale marine heatwaves in the NEP region, identified here and previously as 'warm blobs', have attracted wide attention since the long-lived, so-called 'Blob' event of 2013–2016[1–3] and another event in 2019–2020[4–6]. Since the probability of such marine heatwaves is expected to increase with global warming[7,8], it is critical to understand their driving mechanisms to better understand warm blob potential predictability[9]. Both atmospheric (e.g., anomalous atmospheric pressure ridge and clouds) and oceanic (e.g., salinity-related mixed layer depth variation) processes have been suggested to play important roles in the warm blob formation[1,2,10–12].

Among these key processes, a strong and persistent higher-than-normal sea level pressure (SLP)[1] over the NEP is thought to be critical by modulating surface latent heat flux as well as wind-driven ocean currents and mixing in winter[1,10]. A number of studies suggest that atmospheric teleconnections originating from the tropics can trigger this anomalous pressure ridge over the NEP[2,13–18]. For example, Wang et al.[16] proposed a cross-Pacific pathway of Rossby wave energy at 850 hPa during winter 2013/14 in response to anomalous convection over the Philippine Sea, which contributed to the anomalous ridge over the Gulf of Alaska. Shi et al.[18] discussed potential teleconnections to the NEP triggered by different types of El Niño events. However, unlike these important processes originating in the tropics, possible contributions from midlatitudes to the anomalous ridge have received much less attention.

Rossby wave trains in the mid-latitudes, accompanied by jet stream variations, play essential roles in generating surface air temperature variations in boreal autumn and winter[19–24]. Moreover, the importance of Rossby wave-induced effects has increased during the past two decades[25]. Here we will demonstrate the importance of extratropical wave trains, specifically those originating from the North

[1]Frontier Science Center for Deep Ocean Multispheres and Earth System (FDOMES) and Physical Oceanography Laboratory, Ocean University of China, Qingdao, China. [2]College of Oceanic and Atmospheric Sciences, Ocean University of China, Qingdao, China. [3]Laoshan Laboratory, Qingdao, China. [4]Department of Earth and Planetary Sciences, Yale University, New Haven, USA. [5]LOCEAN/IPSL, Sorbonne University, Paris, France. [6]Institute for Marine and Antarctic Studies, University of Tasmania, Hobart, Tasmania, Australia. [7]Australian Research Council Centre of Excellence for Climate Extremes, University of Tasmania, Hobart, Tasmania, Australia. [8]State Key Laboratory of Earth Surface Processes and Resource Ecology, Beijing Normal University, Beijing, China. ✉e-mail: zhangyu@ouc.edu.cn; huangf@ouc.edu.cn

Atlantic (NATL) and Mediterranean Sea regions, on the long-lasting ridge over the NEP, thus providing insights into the role of remote teleconnections in maintaining the NEP ridge.

## Results

Using data for 13 warm blob events that peak during the boreal cold season (from November to March; Supplementary Table 1 and Fig. 1)[5,26], we firstly consider the composite evolution of geopotential height anomalies at 500 hPa from September to the following February (Fig. 1). An anomalous ridge over the NEP emerges in September (Fig. 1a), somewhat weakens and shifts westward in October (Fig. 1b), reaches its maximum intensity in November (Fig. 1c), and weakens and moves northward in the following months (Fig. 1d–f). This ridge has a quasi-barotropic structure and extends from the surface to the upper troposphere (Supplementary Fig. 2). Aside from the ridge, a trough centered north of the Great Lakes deepens (Fig. 1 and Supplementary Fig. 2), entraining cold air over North America[14,16]. In November, prominent geopotential height anomalies are also found over the eastern NATL, featured by a meridional dipole (Supplementary Fig. 2). This atmospheric pattern may be related to the North Atlantic Oscillation (NAO), with 9 out of 13 warm blob events (~70%) occurring during the positive NAO phase. On the planetary scale, such ridge and trough structures resemble the tropical Northern Hemisphere pattern[27] in winter (Fig. 1d, e), which is potentially important for the warm blobs[10]. To elucidate the intensification of the ridge during the lifetime of the warm blobs, we mainly focus on the results in November hereafter due to the strongest intensity of the anomalous ridge (Fig. 1c).

To explore the potential relationship between the anomalous ridge and midlatitude wave trains, we analyze meridional wind anomalies (shading in Fig. 2a) at 300 hPa in November for warm blobs peaking during the cold season. Two prominent wave trains appear to originate over the NATL, possibly associated with the NAO. One propagates along the subtropical jet stream with a larger wave number and another is along the subpolar jet stream with a smaller wave number. The jet streams act as waveguides for the transportation of energy downstream. The two wave trains converge near Japan, intensifying the wave amplitude and further propagating towards the NEP.

Rossby wave energy, indicated by the wave activity flux (WAF; "Methods"), greatly intensifies over the Mediterranean Sea (vectors in Fig. 2a), implying a potential Rossby wave source (RWS) in this region. Although the two wave trains somewhat weaken over the Eurasian continent, they converge and strengthen again near Japan and travel northeastward to the NEP, which appears to be important for the development and persistence of the anomalous ridge above the warm blobs (Fig. 1 and Supplementary Fig. 2).

For comparison, meridional wind anomalies and WAF are also shown for October (Supplementary Fig. 3a). Although the wave train structure is clear, it is much weaker compared to that in November (Fig. 2a). Consistently, the WAF is much weaker, especially over the Mediterranean Sea and Europe (Supplementary Fig. 3a). Note that the NATL and Mediterranean regions originated circumnavigating wave trains were not reported in Wang et al.[16], but rather they emphasized the wave origin near Japan. Although similar wave trains have been documented in previous studies[23,28–31], none of these focus on the wave propagation towards the NEP. Nevertheless, a recent study implied that the subtropical wave train is closely connected to intense Rossby wave activity over the Mediterranean region and anomalous warming over the NEP[28].

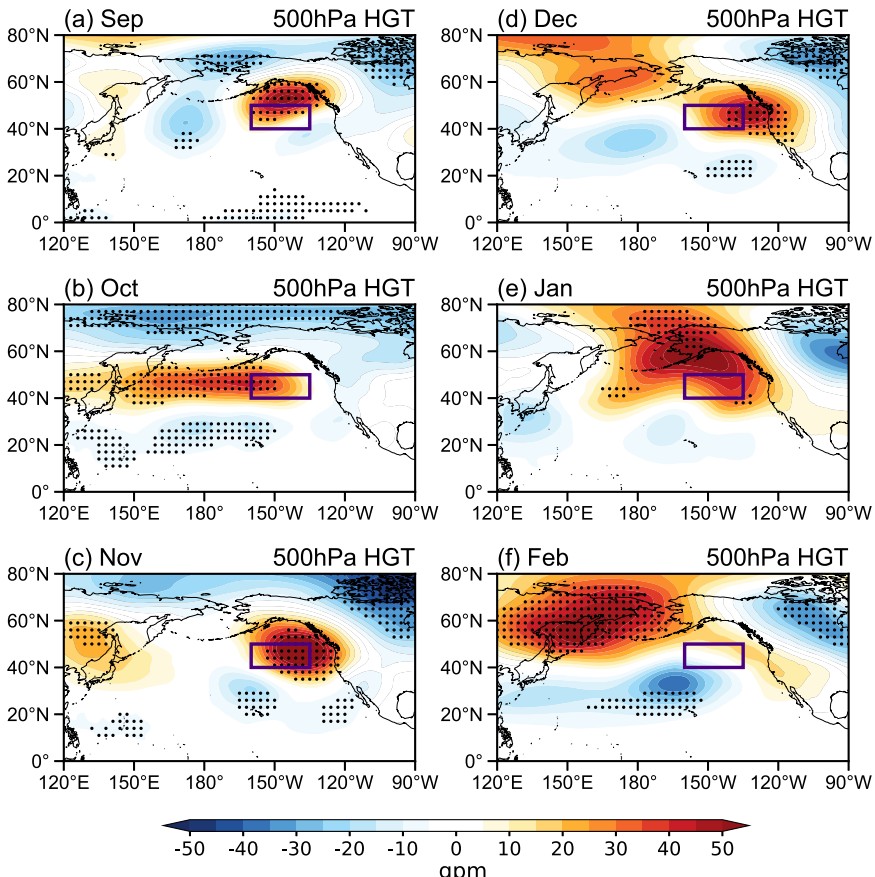

**Fig. 1 | Geopotential height anomalies at 500 hPa associated with warm blobs.** **a–f** Composite geopotential height anomalies (shading, in gpm) at 500 hPa from September through February. Stippling indicates exceeding a 0.1 significance level based on the two-tailed Student's t-test. Warm blob events used for computations are listed in Supplementary Table 1. Purple boxes represent the study area for the warm blobs.

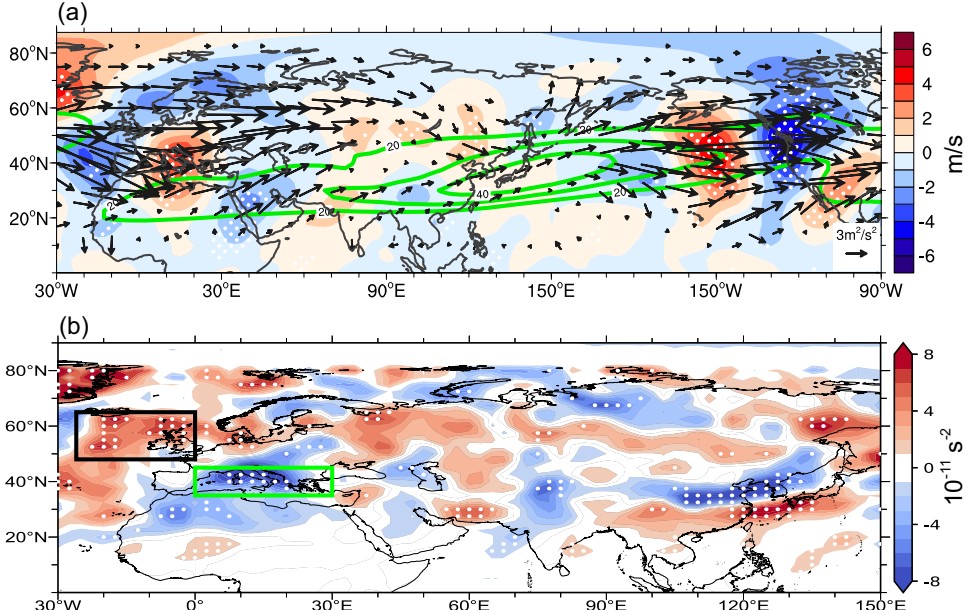

**Fig. 2 | Rossby wave trains and their source for warm blobs peaking during the cold season. a** Composite meridional wind anomalies (shading, in m/s), mean zonal winds (green contour, in m/s), and WAF (vectors, in m²/s²) at 300 hPa. **b** RWS (shading, in 10⁻¹¹/s²) at 300 hPa. Stippling in panels (**a**) and (**b**) indicates meridional wind anomalies and RWS exceeding a 0.1 significance level based on the two-tailed Student's *t*-test, respectively. The green and black boxes in (**b**) outline the Mediterranean region (35°N–45°N, 0°–30°E) and the NATL (48°N–65°N, 26°W–0°), respectively. WAF wave activity flux, RWS Rossby wave source.

Rossby waves are usually triggered when strong convection (or rainfall) generates anomalous condensation (latent) heating, inducing anomalous flow divergence and vorticity in the upper troposphere[32,33], which often occurs over tropical oceans[34–38] and the NATL[39–41]. East of the NATL, Rossby wave trains often bifurcate into two branches over the Mediterranean[28], acting as a pivotal region for Rossby wave propagation and generation[29,42]. The reasons for this wave train bifurcation are: (1) the wave train near the Mediterranean is located outside of the area with strong absolute vorticity gradient so that it is partially reflected[28], and (2) the wave train is sensitive to the origin of the Rossby wave[30].

To uncover the origin of the Rossby wave train that contributes to the NEP warm blobs, we calculate the RWS ("Methods"; shading in Fig. 2b) at 300 hPa in November for warm blobs peaking in the cold season. A large area with a positive RWS is identified over the midlatitude eastern NATL while negative RWS values are detected over the Mediterranean region[23,43]. The two RWS areas correspond to anomalous convergence and divergence in the upper troposphere, respectively (not shown). We mainly focus here on these two RWS regions due to their stronger WAF. In addition, a band of positive RWS is found from southern China to south of Japan, which is consistent with the WAF trajectory along the subtropical jet (Fig. 2a). Further north, another area with a negative RWS is also significant over northern China (Fig. 2b).

In contrast, the RWSs over the NATL and Mediterranean regions are much weaker in October (Supplementary Fig. 3b) compared to November (Fig. 2b), in accordance with the weaker WAF (Supplementary Fig. 3a). Two other RWS areas over China also change greatly in location (Supplementary Fig. 3b). Although there are other centers of RWS, they are variable in time or cover a relatively small area (Fig. 2b and Supplementary Fig. 3b). Further, since the RWS accounts for the location of Rossby wave emanation induced by anomalous vertical motion, upper-tropospheric divergence, and vorticity[32], we conjecture that the above-mentioned major RWS regions are potentially related to local rainfall anomalies.

To assess the rainfall variations associated with the NEP warm blobs, we estimated the rainfall anomalies for warm blobs peaking during the cold season (Fig. 3a). In November, significant negative rainfall anomalies are identified over the eastern NATL, in association with the NAO-related positive geopotential height anomalies (Supplementary Fig. 2). Moreover, we find significantly enhanced rainfall over the Mediterranean region (Fig. 3a). In fact, 10 out of the 13 warm blob events (~77%) correspond with higher-than-normal rainfall over the Mediterranean region (for example, the 2013/14, 2015, and 2019 events). To better understand the rainfall anomalies over the Mediterranean region, we show both the monthly rainfall climatology and the variability (Supplementary Fig. 4). Overall, this region has abundant rainfall from October to February, with a maximum of around 90 mm in November. However, a relatively limited amount of precipitation falls from June to August. These distinct seasonal differences are typical for the Mediterranean climate. In terms of variability, year-to-year rainfall variations are large between November and February. Rainfall anomalies in October are much weaker, especially over the Mediterranean region (Supplementary Fig. 5a), again suggesting that the Mediterranean region may play an important role in sustaining the wave train and anomalous ridge over the NEP in November. In addition, a meridional rainfall dipole (Fig. 3a) associated with the two RWS centers (Fig. 2b) is found over China and surrounding seas.

According to previous studies[32,33], anomalous diabatic heating sustained by rainfall/convection is essential for exciting atmospheric Rossby waves. Hence, we show a composite apparent heat source ("Methods") at 300 hPa for warm blobs peaking during the cold season (Fig. 3b). Prominent heating and cooling anomalies are found over the Mediterranean and NATL regions, respectively, consistent with the rainfall anomalies (Fig. 3a). However, the two rainfall and RWS centers over China and surrounding seas are not robust in terms of anomalous heating (Fig. 3b).

Further, to verify the roles of the Mediterranean and NATL regions in triggering the wave trains that contribute to the long-lasting anomalous ridge over the NEP, we impose anomalous heating/cooling in November within an atmospheric linear baroclinic model (LBM;

"Methods")[44]. Background atmospheric conditions in November given in the LBM are presented in Supplementary Fig. 6. The vertical structure of the apparent heat source over the Mediterranean region (black box in Fig. 3) exhibits anomalous heating induced by the positive rainfall anomalies throughout the whole troposphere, with a maximum magnitude at 300 hPa (Fig. 4a). After the simulation reaches a quasi-equilibrium state, a wave train originating from the Mediterranean region is identified in the extratropics with the WAF propagating eastward to the NEP (Fig. 4b). Over the NEP (purple box in Fig. 4b), an anomalous ridge is evident at 500 hPa (Fig. 4b). At the surface, the anomalous high-pressure system shifts a bit northwestward (Fig. 4c), with easterly anomalies blowing against climatological midlatitude southwesterlies (Supplementary Fig. 6c), which are favorable for inducing the NEP warm blobs by reducing latent heat loss from the ocean to the atmosphere[1,5,26].

In terms of the detailed evolution, after imposing heating over the Mediterranean region, positive geopotential height anomalies appear over the NEP approximately 16 days after the heating and then develop and become stable in the following days (Supplementary Fig. 7). Moreover, Rossby wave energy can disperse further downstream from the anomalous ridge, amplifying the trough over North America and leading to a displacement of the polar vortex[16]. In addition, when the anomalous heating is centered at 700 hPa[34], the positive geopotential height anomalies are weaker and shift westward (Supplementary Fig. 8). Hence, the LBM experiment reproduces the anomalous ridge reasonably well over the NEP forced by anomalous heating over the Mediterranean region.

Further examining the role of the NATL forcing, we impose cold anomalies over the NATL (Fig. 4d). This results in positive geopotential height anomalies at 500 hPa over the NEP (Fig. 4e), which resemble the observations very well (Fig. 1c). However, surface westerly wind anomalies associated with a cyclonic circulation anomaly in the study area are not favorable for the warm blobs (Fig. 4f). The difference between circulation anomalies at 500 hPa and those at the surface may arise from the baroclinic response of the LBM.

On the other hand, we also notice that the WAF converges near East Asia (Fig. 2a), which might further boost the wave train. To test this possibility, we impose anomalous cooling/heating according to the structure of the apparent heat source over this region (Supplementary Figs. 10 and 11). Although a weak anomalous ridge emerges in the heating experiment (Supplementary Fig. 11), the heating (Fig. 3b) in these two regions is scattered and not robust. In this study, we have mainly explored the roles of the Mediterranean and NATL regions, which appear to be the origin of extratropical teleconnections, rather than a midway booster, that sustain the anomalous ridge.

Next, to confirm the driving role of the Mediterranean and NATL regions in exciting the wave trains and anomalous ridge over the NEP, we conduct atmospheric model experiments using the sophisticated CAM5 (i.e., version 5.0 of the Community Atmosphere Model; "Methods"). Results show positive geopotential height anomalies over the NEP and subpolar North America in October (Fig. 5b) and November (Fig. 5c) when the anomalous warming pattern is imposed in the Mediterranean region (Fig. 5a). The location of the modeled anomalous ridge is shifted eastward compared to the observations (Fig. 1). When the sea surface temperature (SST) warming pattern associated with the largest enhanced rainfall in the Mediterranean region is prescribed, the anomalous ridge over the NEP is stronger, with a similar location to the observations (left column of Supplementary Fig. 12), suggesting that the Mediterranean region can play an important role in driving this anomalous ridge. When composite SST anomalies during the warm blobs are superimposed onto the climatology in the NATL (Fig. 5d), the anomalous ridge is robust in November (Fig. 5f). In the Exp_NAtl_NAO experiment (right column of Supplementary Fig. 12), triple-like SST anomalies with a broad cooling in the eastern NATL are

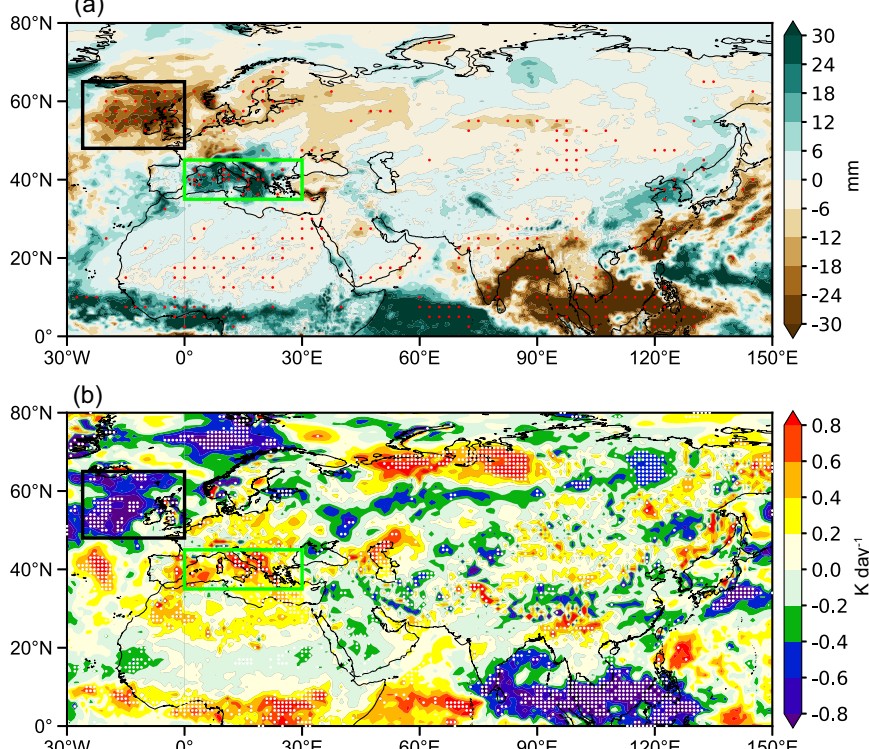

**Fig. 3 | Rainfall and diabatic heating anomalies in November for warm blobs peaking during the cold season.** Composite (**a**) rainfall (shading, in mm) and **b** 300-hPa diabatic heating ($q_1$; shading, in K/day) anomalies. Stippling indicates exceeding a 0.1 significance level based on the two-tailed Student's $t$-test. The green and black boxes mark the Mediterranean and North Atlantic, respectively.

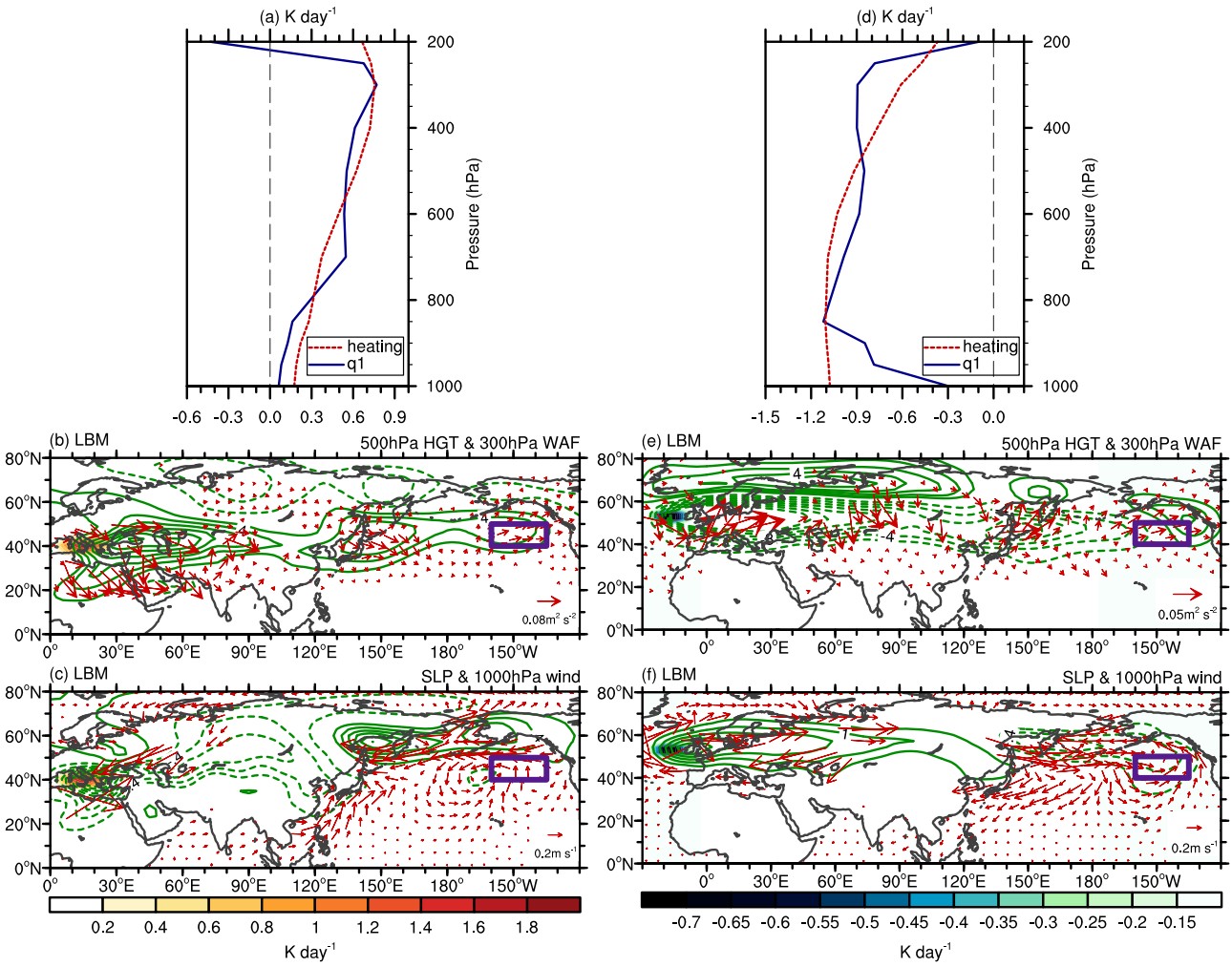

**Fig. 4 | Steady atmospheric circulation response to anomalous heating/cooling over the Mediterranean and NATL regions in the LBM. a, d** Vertical profiles of $q_1$ anomaly (blue line) and the imposed heating (red line) in the LBM over the Mediterranean and NATL regions, respectively. Note that the heating was multiplied by a factor of two to obtain a more robust response. **b** Geopotential height anomalies (contours, in gpm) at 500 hPa and WAF (vectors, in m²/s²) at 300 hPa and **c** SLP anomalies (contours, in hPa) and surface winds anomalies (vectors, in m/s) in the Mediterranean experiment. Shading indicates the imposed heating. Purple boxes represent the study area for the warm blobs. **e, f** Same as (**b, c**), but for the NATL experiment. HGT geopotential height, NATL North Atlantic, LBM linear baroclinic model, WAF wave activity flux, SLP sea level pressure.

prescribed, associated with the positive phase of the NAO. A prominent anomalous ridge over the NEP and polar regions is found in October and November, resembling the observations (Fig. 1). Thus, the effects of both the Mediterranean and NATL regions on the NEP anomalous ridge are captured by the CAM5 simulations.

## Discussion

In summary, this study provides insights into the development of the anomalous NEP ridge from the perspective of extratropical wave trains, highlighting downstream climate impacts reaching the NEP from the Mediterranean and NATL regions. The wave train dynamics could provide a potentially important source of predictability for the anomalous ridge and the resultant cold-season NEP warm blobs as well as North American temperature anomalies. We emphasize that this mechanism prominent in November may not be applicable for other winter months because the background state plays a crucial role in the generation and guidance of Rossby waves, influencing the establishment of teleconnection patterns. Moreover, the contribution from the tropics should not be neglected, as it could be amplified when combined with extratropical forcing (e.g., from the Mediterranean; Supplementary Fig. 13).

## Methods

### Observational and reanalysis datasets

For SST, we use monthly data from the National Oceanic and Atmospheric Administration (NOAA) Extended Reconstructed SST version 5 (ERSST v5), gridded at 2° × 2°[45]. For precipitation, wind, SLP, temperature, and geopotential height, we use the European Centre for Medium-Range Weather Forecasts (ECMWF) ERA5 reanalysis[46]. The horizontal resolution of ERA5 data is 0.25° × 0.25°. The data prior to 1959 are from an ERA5 preliminary version[47]. We also use velocity potential, divergence, and relative vorticity data from the Japanese 55-year Reanalysis (JRA-55) at 2.5° × 2.5° resolution[48]. We analyze the period from 1951 to 2021 in this study. Anomalies of variables are calculated relative to a baseline climatology from 1979 to 2021.

### Composite analysis of the NEP warm blobs

The study area of the warm blobs is 40°N–50°N, 160°W–135°W. Following the definition from previous studies[5,26], a warm blob event is identified when normalized monthly SST anomalies averaged over the study area (i.e., the blob index) are greater than 1.0 for no fewer than 5 months with at most 1-month interruption. Accordingly, 13 warm blob events (Supplementary Table 1) peaking during the cold season

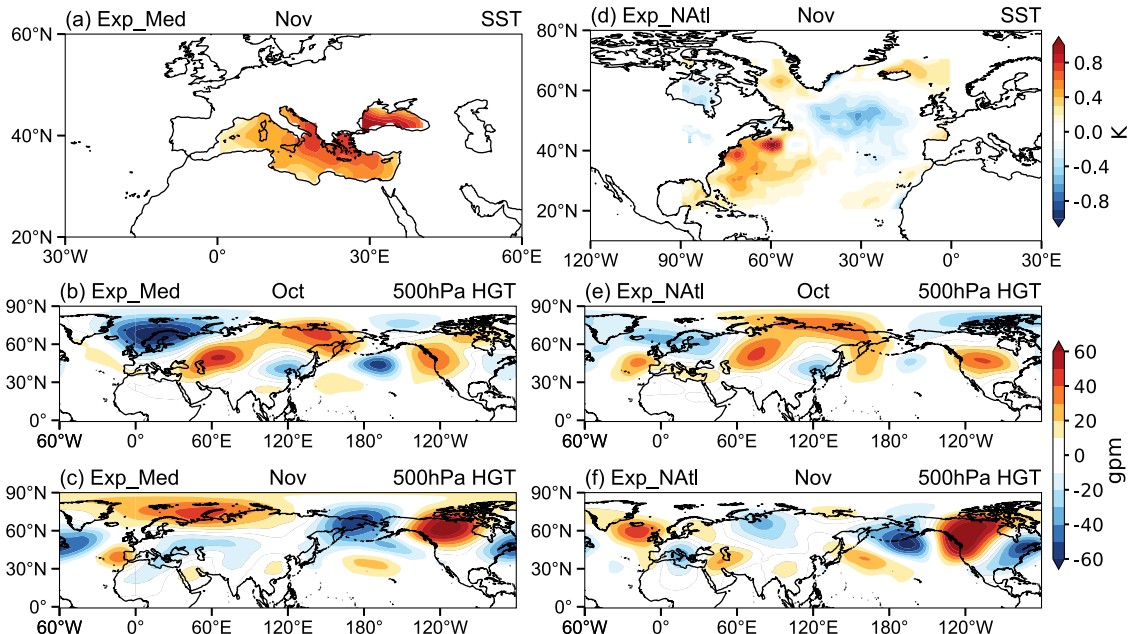

**Fig. 5 | Response of geopotential height to the prescribed SST forcing in CAM5.** **a**, **d** Prescribed SST anomalies (shading, in °C) relative to climatology in Exp_Med and Exp_NAtl, respectively. **b**, **c** Geopotential height anomalies (shading, in gpm) at 300 hPa in Exp_Med relative to CTRL in October and November. **e**, **f** As in (**b**, **c**), but for Exp_NAtl. SST sea surface temperature, CAM5 Community Atmosphere Model 5.0.

(from November to March) are selected to perform our composite analysis. The blob index is calculated after removing linear trends[5,26]; such an approach is widely used, although the SST variations do have nonlinear features[49–52]. Peak intensity is given by the blob index when reaching the maximum. The return period is computed according to refs. 7,53. Note that due to the period of the JRA-55 dataset, we use 11 warm blob events after 1958 when calculating the RWS.

**Calculation of WAF**

To analyze the propagation of Rossby wave energy in the Northern Hemisphere, the horizontal WAF is computed following Takaya and Nakamura[54]:

$$\mathbf{W} = \frac{P\cos\phi}{2|\mathbf{U}|} \cdot \begin{pmatrix} \frac{U}{a^2\cos^2\phi}\left[\left(\frac{\partial\psi'}{\partial\lambda}\right)^2 - \psi'\frac{\partial^2\psi'}{\partial\lambda^2}\right] \\ + \frac{V}{a^2\cos\phi}\left[\frac{\partial\psi'}{\partial\lambda}\frac{\partial\psi'}{\partial\phi} - \psi'\frac{\partial^2\psi'}{\partial\lambda\partial\phi}\right], \\ \frac{U}{a^2\cos\phi}\left[\frac{\partial\psi'}{\partial\lambda}\frac{\partial\psi'}{\partial\phi} - \psi'\frac{\partial^2\psi'}{\partial\lambda\partial\phi}\right] \\ + \frac{V}{a^2}\left[\left(\frac{\partial\psi'}{\partial\phi}\right)^2 - \psi'\frac{\partial^2\psi'}{\partial\phi^2}\right] \end{pmatrix} \quad (1)$$

where $\mathbf{W}$ (unit: m² s⁻²) denotes the horizontal WAF; $P$ is pressure/(1000 hPa); $\mathbf{U}$ (unit: m s⁻¹) represents the basic flow with the zonal component $U$ and meridional component $V$; $\Psi(=\Phi/f)$ is the stream function; $a$ is the radius of the Earth; $\phi$ and $\lambda$ denote latitude and longitude, respectively; $\Phi$ is geopotential height, and $f = 2\Omega\sin\phi$ is the Coriolis parameter; where $\Omega$ is the rotation rate of the Earth. Primes indicate anomalies relative to climatology.

**Calculation of RWS**

To locate the regions that can trigger Rossby waves in the upper troposphere, the Rossby wave source (RWS) is calculated at 300 hPa following Sardeshmukh and Hoskins[32]:

$$RWS = -\nabla_H \cdot \left\{\boldsymbol{u}_\chi(f+\zeta)\right\}' \quad (2)$$

where $\mathbf{u} = (u, v)$ denotes the horizontal wind velocity vector and subscript $\chi$ indicates its divergent component. $\zeta$ and $f$ are relative vorticity and planetary vorticity, respectively; $\nabla_H$ is the horizontal gradient operator. Primes indicate anomalies.

**Calculation of the apparent heat source**

To isolate the anomalous heating and cooling pattern associated with precipitation anomalies, we calculate the atmospheric diabatic heating ($q_1$) as below[55] based on data from the ERA5 dataset:

$$q_1 = \frac{\partial T}{\partial t} - (\omega\sigma - \boldsymbol{V} \cdot \nabla_H T) \quad (3)$$

where $T$ is air temperature; and $\omega$ is the vertical velocity in pressure coordinates. Static stability is expressed as $\sigma = (RT/c_p p) - (\partial T/\partial p)$, where $c_p$ is the specific heat capacity of air at constant pressure; $R$ denotes the specific gas constant, and $p$ is pressure. $\boldsymbol{V} = (u, v)$ is the horizontal wind vector, and $\nabla$ is the horizontal gradient operator. Here, $q_1$ represents total diabatic heating, including radiation, latent and sensible heat fluxes, as well as subgrid-scale heat flux convergence[55].

**Experimental setup in the LBM**

The atmospheric linear baroclinic model (LBM) is widely used to explore linear atmospheric dynamics[44]. The LBM is based on a linearized version of the primitive equations. Given a basic state $\bar{X}$, a steady response $X$ follows an equation written in a matrix form as

$$L(\bar{X})X = \mathbf{F} \quad (4)$$

where $\mathbf{F}$ indicates a forcing vector, and $L$ is a linear dynamical operator related to the perturbed primitive equations[44]. In this study, the LBM is utilized to examine whether prescribing heating/cooling over the Mediterranean and NATL can drive a Rossby wave train that sustains an atmospheric ridge over the NEP in accordance with the observations. The model is run at T42L20, with a nominal horizontal resolution of 2.8° × 2.8° and 20 vertical sigma levels. The model consists of basic

equations linearized with respect to the atmospheric mean state in November derived from the NCEP reanalysis[56]. Dissipation in the model includes (1) a bi-harmonic horizontal diffusion with a damping timescale of 1 day for the shortest waves, (2) a weak vertical diffusion (a damping timescale of 1000 days) to remove noise arising from finite differencing, (3) Newtonian damping, and (4) Rayleigh friction represented by a linear drag, with a timescale of 1 day applied only to the lower boundary layers and the uppermost two levels[57]. The simulations are run for 20 days to reach a quasi-equilibrium atmospheric state[57–59]. Output variables, including geopotential height, winds, and sea level pressure, are averaged over days 16–18 of the simulations, which yields a steady response to the prescribed heating/cooling. Note that the imposed heating/cooling in the Mediterranean and NATL experiments is multiplied by a factor of two to obtain a robust response as we calculate mean values over these regions.

### Experimental setup in CAM5

CAM5 model was developed by NCAR. Five experiments were conducted: a control experiment (CTRL), two Mediterranean experiments (Exp_Med and Exp_Med_rain), and two NATL experiments (Exp_NAtl and Exp_NAtl_NAO; Supplementary Table 2). We use the f19_g16 model configuration (i.e., 2.5° × 1.9° atmospheric resolution) for the experiments. The prescribed SST[60] anomalies are computed from the merged products of HadISST version 1[61] and NOAA OISST version 2[62]. The CTRL run is forced by monthly climatological SST globally over 1981–2010. For the Exp_Med run, composite SST anomalies during the warm blobs are superimposed onto climatological SST in the Mediterranean region (Fig. 5a). For the Exp_Med_rain run, SST anomalies averaged in 2013 and 2019 (when the Mediterranean has the largest enhanced rainfall) are superimposed onto climatological SST in the Mediterranean region (Supplementary Fig. 10a). For the Exp_NAtl run, composite SST anomalies during the warm blobs are superimposed onto climatological SST in the NATL (Fig. 5d). For the Exp_NAtl_NAO run, SST anomalies in 1993 with the largest NAO are superimposed onto climatological SST in the NATL (Supplementary Fig. 10d). For these sensitivity experiments, SST climatology is prescribed in the regions outside of the above forcing area. Each experiment is initialized from January 1, 1979 and integrated for 25 years, with the last 20 years averaged for our analyses.

## Data availability

The ERSST v5 data are available at https://psl.noaa.gov/data/gridded/data.noaa.ersst.v5.html. The ERA5 data can be accessed at https://cds.climate.copernicus.eu/cdsapp#!/dataset/reanalysis-era5-pressure-levels-monthly-means?tab=form, https://cds.climate.copernicus.eu/cdsapp#!/dataset/reanalysis-era5-single-levels-monthly-means?tab=overview, https://cds.climate.copernicus.eu/cdsapp#!/dataset/reanalysis-era5-pressure-levels-monthly-means-preliminary-back-extension?tab=overview, and https://cds.climate.copernicus.eu/cdsapp#!/dataset/reanalysis-era5-single-levels-monthly-means-preliminary-back-extension?tab=overview. The JRA-55 data are available at https://rda.ucar.edu/datasets/ds628.1/index.html#cgi-bin/datasets/getWebList?dsnum=628.1&action=customize&disp=. The model output and source data used in this study are available on Zenodo at https://zenodo.org/doi/10.5281/zenodo.10473150.

## Code availability

This article uses Python (https://anaconda.org/) and NCL to perform the reported analyses. The codes that support the findings of this study are available at https://doi.org/10.5281/zenodo.10494745.

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

## Acknowledgements

This work is supported by the National Natural Science Foundation of China (42006013, 42230405, and 42276006 to J.S.; 42075024 to F.H.), Laoshan Laboratory (No. LSKJ202202503 to J.S.; No. LSKJ202202602 to Y.Z.), the National Key Research and Development Program of China (2019YFA0607004 to F.H.), China Postdoctoral Science Foundation (2021M703034 to Y.Z.), the ARC Centre of Excellence for Climate Extremes (CE170100023 to N.J.H.), NASA (80NSSC21K0558 to A.V.F.), NOAA (NA20OAR4310377 to A.V.F.), and the ARCHANGE project (ANR-18-MPGA-0001, France to A.V.F.). We greatly thank Marine Big Data Center of the Institute for Advanced Ocean Study of Ocean University of China for providing computational resources. We thank Prof. Shang-Ping Xie for his insightful comments and edits for this paper. We also thank Drs. Hong Wang, Boqi Liu, Zedong Liu, Peng Liu, Xiadong An, and Tengfei Yu for their helpful suggestions and comments on model set-up, experimental design, and result interpretation.

## Author contributions

J.S. conceived the study and wrote the manuscript. H.H. plotted the figures and conducted the analyses. F.H., A.V.F., N.J.H., and Y.Z. provided additional thoughts and helped revise the manuscript. Y.L., J.C., and X.H. helped improve the figures. R.D., S.W., Q.L., and X.L. were involved in interpreting the results.

## Competing interests

The authors declare no competing interests.
