## [Peer Review File · Nature Communications]

Northeast Pacific warm blobs sustained via extratropical atmospheric teleconnectionsReviewers' Comments:

Reviewer #1 (Remarks to the Author):

The study titled "Teleconnection from heating by Mediterranean rainfall sustains the atmospheric ridge over the Northeast Pacific warm blob" utilized observations/reanalysis datasets and linear baroclinic model simulations to examine the teleconnection effects on the Northeast Pacific warm blob. The findings are intriguing and may contribute to our comprehension of the mechanisms responsible for the formation of the Northeast Pacific blob. However, upon reviewing the study, I have identified a few concerns that could introduce ambiguity in presenting the results and understanding the underlying mechanism. Please find below my major and minor comments.

Major comments:

- 1. The definition of blob.** The definition of the blob used in this study could benefit from improvement or further clarification. Firstly, while the study area is limited to 40°N-50°N and 160°W-135°W, the evolution of the blob, such as the 2014-2015 event, extends beyond the defined box of interest. Warm sea water can deform and extend towards coastal regions outside the designated area. Additionally, the vertical structure of the blob has been shown to be somewhat significant in generating extreme heating. Therefore, it raises the question of whether there are more suitable three-dimensional tracking algorithms available to identify each individual blob event and better characterize its spatial evolution.

Secondly, regarding the intensity of the blob, the authors considered normalized monthly sea surface temperature (SST) anomalies with box-averaged values greater than 1.0. It would be helpful to know the standard deviation for each identified blob event in this study. Are the most recent events characterized by significantly larger standard deviation values compared to other events? If so, it suggests that the formation mechanisms and preconditions for strong blobs may differ considerably from those of weaker blobs, and perhaps being modified by global warming. Furthermore, it would be informative to understand where these blobs lie within the full distribution of SST anomalies in the specified regions and their return period. Providing these basic statistical analyses could enhance the illustration of extreme events in this region.

Overall, addressing these points would contribute to a clearer understanding of the spatial and intensity characteristics of the blob events under investigation in this study.

- 2. Atmospheric teleconnection in different month.** The atmospheric teleconnection is derived from the analysis of winter blobs, which occur between November and March (Extended Data Table 1), as well as the preceding months. The authors' perspective is that this teleconnection is a result of stationary Rossby wave trains (Figures 2 and 4). In light of this viewpoint, it becomes important to carefully evaluate the impact of different background winds during various months on the atmospheric teleconnection. Background fields play a crucial role in the generation and guidance of Rossby waves, consequently influencing the establishment of distinct teleconnection patterns (e.g., Hoskins and Woollings 2015). Following the logic of examining prior months with lead

times of 3, 2, 1, and zero, there is a concern that the teleconnection pattern observed between November and February may differ significantly from the pattern observed between August and November. This disparity could also lead to inconsistencies in the mean state of the atmosphere, as utilized in the LBM simulations, which will be further discussed in the subsequent comment.

The issue of causality in the study poses some challenges in interpretation. In Figure 1, the authors aimed to employ composite analysis with a lead month to provide insights into causality. However, two questions arise from this approach. Firstly, it is unclear why the strongest geopotential height anomalies do not occur at the peak time but rather at a lead of 1 month. Secondly, how can the authors be confident that there is not a third-party or confounding factor simultaneously driving both the blob and teleconnection phenomena? Lead-lag composite or correlation analyses alone may not be sufficient to reject such a possibility. Furthermore, Figure 2a presents composite meridional wind anomalies and WAF, but it remains questionable whether the WAF is causing the wind anomalies or vice versa, or by a confounding factor. We could also rationalize that the winds anomalies in the Mediterranean region are a response to upstream WAF emanating from the North Atlantic. The utilization of the Granger causality algorithm (e.g., Silva et al. in 2021) could potentially provide further insights in this regard, although I am not sure the authors have time or efforts to carry out the analysis in this study.

The authors' inclusion of LBM simulations serves the purpose of addressing the causality issue. However, it is important to note that LBM simulations may also have their limitations, as described in the subsequent comment.

- 3. The design of IBM simulations.** In addition to the concerns regarding the background wind issue mentioned earlier, it is worth noting that the LBM used in this study appears to be a simplified representation of atmospheric dynamics. It is important to acknowledge that there is a significant disparity between the LBM and state-of-the-art global climate models (GCMs), such as CESM2, which are commonly utilized to investigate teleconnections and associated dynamics. Considering this disparity, it would be valuable for the authors to consider conducting GCM simulations to complement and support their findings based on observations and reanalysis data, as well as LBM simulations. By employing a GCM, they could potentially enhance the robustness of their conclusions by capturing a broader range of atmospheric processes and interactions. GCM simulations could provide a more comprehensive and realistic representation of the teleconnection phenomenon under investigation.

To further address the causality issue, it could be beneficial to conduct a series of LBM simulations (and potentially GCM simulations) with forcings specifically applied to hotspots in southeast China, the Ural Mountain region, or the North Atlantic. By doing so, the authors can examine whether a similar teleconnection pattern emerges, as indicated by the statistically significant RWS presented in Figure 2b. These targeted simulations would provide an opportunity to assess the robustness of the observed teleconnection

pattern and its relationship to the specified forcings. If inconsistent teleconnection patterns are observed across multiple simulations with different forcings applied to these specific regions, it would lend support to the hypothesis that the Mediterranean regions play a causal role in driving the teleconnections. This approach would strengthen the authors' argument regarding the causality between the identified hotspots and the observed teleconnection pattern.

Minor comments:

Line 46: What is salinity variation? Can the authors discuss more about its impact on blob?

Line 65: Are all blobs peaking in boreal winter? I saw in the Extended Data Table 1 some blobs peak in March and November, which may not be typical boreal winter months: December, January, and February. How about using cold season in the Northern Hemisphere?

Line 74: do you use TNH abbreviation in the following part of manuscript?

Lines 76-77: do the authors mean wintertime midlatitude wave trains? Or autumn wave trains?

Lines 76-83 and 84-86: Why the wave train, originating in the North Atlantic, does not coincide with the WAF, from Mediterranean region?

Lines 99-100: A recent study (Hong and Hsu 2023) showed that the central Pacific could affect Northeast Pacific. How do the authors reconcile their results and those presented in this study?

Lines 108-110: why the wave trains often bifurcate into two branches over the Mediterranean? Can the authors discuss more?

Lines 136-137: can the authors show their evolution?

Lines 154-155: Does the strongest magnitude happen at 300 hPa because of the vertical resolution? If the vertical resolution increases, could the magnitude shift to, for example, 309 hPa?

Lines 158-159: what are the background mean fields given in the LBM simulation? Can the authors show their spatial maps?

Line 231: As the authors had calculated three-dimension WAF, I am curious what is the role of vertical component here?

Line 268: what does the gamma-type mean?

Line 455, Figure 1L since the authors mentioning TNH pattern, which has anomaly centers in the North Atlantic, can you extend the domain to the North Atlantic?

Extended Data Table 1: the bottom two events are the same, except “2019” and “2020”. Can the authors clarify?

References

Hong, H.J. and Hsu, H.H., 2023. Remote tropical central Pacific influence on driving sea surface temperature variability in the Northeast Pacific. *Environmental Research Letters*, 18(4), p.044005.

Hoskins, B. and Woollings, T., 2015. Persistent extratropical regimes and climate extremes. *Current Climate Change Reports*, 1, pp.115-124.

Silva, F.N., Vega-Oliveros, D.A., Yan, X., Flammini, A., Menczer, F., Radicchi, F., Kravitz, B. and Fortunato, S., 2021. Detecting climate teleconnections with Granger causality. *Geophysical Research Letters*, 48(18), p.e2021GL094707.

Reviewer #2 (Remarks to the Author):

This paper looks at the potential role of the Mediterranean region in forcing “warm blobs” in the northeast Pacific by inducing an anomalous atmospheric ridge in that region. The authors claim that enhanced rainfall over the Mediterranean drives a wave train in the northern hemisphere that generates this anomalous ridge, which subsequently leads to marine heatwave events in the North Pacific. The manuscript is well written, but I am unconvinced regarding the causality based on the evidence that has been presented. My main comments are as follows:

Previous studies have linked Northeast Pacific marine heatwaves to the tropics. While the authors reference the link to ENSO and the tropics in the introduction, as far I can tell it is not accounted for in this study nor has the influence of ENSO or the tropics been removed in any way. Therefore I don't think it's possible to rule out the role of the tropics in forcing these extratropical wave trains or at least contributing towards the marine heatwave/anomalous North Pacific anticyclone. Related to this, the rainfall anomaly plots (Fig 3b and extended data fig 4) show a very limited area. It would be useful to see the rainfall anomalies including the rest of the northern hemisphere (including the tropics), to see any other regions of enhanced or suppressed rainfall, as I find it hard to believe that the tropics is not forcing the extratropics in any way. Many areas within the Mediterranean region don't actually have significant rainfall anomalies in most months prior, either, even with the fairly generous significance threshold used (extended data fig 4)

While wave activity flux can be a useful diagnostic, it doesn't give any information about causality. The authors state that “Anomalous Rossby wave energy, indicated by wave activity flux, flows from the Mediterranean region into the subtropical jet”. While there are wave activity flux vectors over the Mediterranean, there are also vectors in most other parts of the extratropics and there is no evidence here to suggest that the waves originate over the Mediterranean itself. The vectors are also scaled in such a way on the plot that they are difficult to distinguish. I also don't think that wave activity flux is particularly good at showing propagation of waves from the tropics to extratropics, which I think is the only evidence the authors use to rule out the role of the tropics. To me, all this panel shows is waves propagating around the northern hemisphere but with no clear information about where they originate or what is forcing them. This is particularly the case when looking at the monthly figures (extended data fig 2)

The Rossby wave source plot (Fig 2b) has many areas of significant source, so why the focus on the small area over the Mediterranean? Many of these areas of RWS will also be generated from tropical rainfall, which doesn't seem to have been taken into account.

It seems to me that what is more likely is that the anomalous ridge over the North Pacific is actually influencing Mediterranean rainfall through perturbing the North Atlantic jet. Again, this doesn't seem to have been considered in the present study. This is also made harder to see by the plots being cut off between 90W-0 longitude. Related to this, there seems to be a break in the wave train over central Asia (Fig 2), which would also support the North Pacific influencing Europe, rather than the other way round.

The authors state that the LBM experiment “closely reproduces the anomalous ridge over the NEP”,

which seems like a bit of a strong statement to me. Yes, there is an anomalous ridge over the North Pacific but it is shifted south somewhat. The wave activity flux is also extremely messy which makes it hard to see what's going on, but this doesn't seem to correspond very closely to Fig 2a either. I suspect that if a similar heating was applied elsewhere in the northern hemisphere (e.g. north of India) that a similar pattern would emerge, so I don't think this provides sufficient evidence of the Mediterranean forcing the North Pacific.

Reviewer #3 (Remarks to the Author):

The paper describes an interesting possible connection between anomalous heating in the Mediterranean region and North Pacific "warm blobs". The results are potentially new, but they need to be better substantiated and supported by more analysis. The paper is well structured, but a few points need to be clarified before being considered for publication.

1. The discussion in lines 76 – 83 need a better motivation. The wavetrain is identified only by two contours and the amplitude over the Mediterranean is minor. The WAF seems to be distributed over a much larger pattern over Europe rather than only over the Med, the RWS has a smaller spatial scale by construction and exhibit many other maxima, for instance over North Europe. In summary, it is hard to conclude that the Med plays any special role.
2. The vertical structure of the heating computed as a residual is surprising (Fig, 7 Extended data and other). At these latitudes is mostly shallow heating (see for instance Ling and Change, 2013), so the profile showing maximum heating at 300mb is rather surprising.
3. The results from the linear model also need to be better justified. The model needs to be described better, especially regarding the dissipation formulation used and the vertical structure of the heating imposed. The cryptic statement at the end of the method section ("A gamma-type ...") needs to be better formulated.
4. Assuming that the solution has converged successfully, the way it develops is suspect. The development of the response in the Pacific can be due to residual amplitude along the equatorial region deriving from the spectral truncation of the localized anomaly prescribed over the Mediterranean. At T42 resolution the expansion of the localized heating over the spectral transform will generate amplitude along the equator that will be picked up by the model in the transform cycle and result in further forcing for the mid-latitudes. This is particularly important in the West Pacific because the basic state is very sensitive to forcing there and it can amplify easily. Depending on the dissipation and cooling imposed this effect may be more or less emphasized, so the modeling choices are very important to clarify.

Reviewers' comments:

Reviewer #1:

The study titled "Teleconnection from heating by Mediterranean rainfall sustains the atmospheric ridge over the Northeast Pacific warm blob" utilized observations/reanalysis datasets and linear baroclinic model simulations to examine the teleconnection effects on the Northeast Pacific warm blob. The findings are intriguing and may contribute to our comprehension of the mechanisms responsible for the formation of the Northeast Pacific blob. However, upon reviewing the study, I have identified a few concerns that could introduce ambiguity in presenting the results and understanding the underlying mechanism. Please find below my major and minor comments.

>>Reply:

We really appreciate your positive and helpful comments that have helped us further improve the manuscript. Throughout our responses below, for convenience, the line numbers in the new revised manuscript are referred to as Revision Lines (RL).

Major comments:

1. The definition of blob. The definition of the blob used in this study could benefit from improvement or further clarification. Firstly, while the study area is limited to 40°N-50°N and 160°W-135°W, the evolution of the blob, such as the 2014-2015 event, extends beyond the defined box of interest. Warm sea water can deform and extend towards coastal regions outside the designated area. Additionally, the vertical structure of the blob has been shown to be somewhat significant in generating extreme heating. Therefore, it raises the question of whether there are more suitable three-dimensional tracking algorithms available to identify each individual blob event and better characterize its spatial evolution.

>>Reply:

Thanks for your valuable comment that may help to identify the warm blobs more suitably. We do agree that the evolution of the warm blobs can extend beyond the defined box of interest. But our previous studies showed that the results are not sensitive to the selection of specific blob area in the Northeast Pacific (Shi et al., 2022; Tang et al., 2021). In addition, although the vertical structure is important for the warm blobs, we do not probe into this three-dimensional aspect in this study to identify the warm blobs. Hence, we used the 13 warm blob events selected in our previous studies (Chen et al., 2021a,b) in this paper.

To make it clearer, we modified the description in RL 253–258.

Secondly, regarding the intensity of the blob, the authors considered normalized monthly sea surface temperature (SST) anomalies with box-averaged values greater than 1.0. It would be helpful to know the standard deviation for each identified blob event in this study. Are the most recent events characterized by significantly larger standard deviation values compared to other events? If so, it suggests that the formation mechanisms and preconditions for strong blobs may differ considerably from those of weaker blobs, and perhaps being modified by global warming. Furthermore, it would be informative to understand where these blobs lie within the full distribution of SST anomalies in the specified regions and their return period. Providing these basic statistical analyses could enhance the illustration of extreme events in this region. Overall, addressing these points would contribute to a clearer understanding of the spatial and intensity characteristics of the blob events under investigation in this study.

>>Reply:

Thanks for this important comment. We have revised the manuscript according to your suggestion. First, we added the standard deviation (i.e., value of blob index) for each identified warm blob event in **Extended Data Table 1**. As we have removed the linear trend when calculating the blob index, the effects of global warming on intensity of the warm blobs are not included in this study. Nevertheless, the warm blob events in the recent decade (i.e., 2010–2020) is stronger than other cases.

Moreover, we added the spatial distribution of SST anomalies of each warm blob event as Extended Data Fig. 1 (also see Fig. R1 below) to provide more information of warm blob events.

Fig. R1 SST anomalies (shading, in °C) at the peak month of each warm blob event. Green boxes represent the study area of the warm blob.

2. Atmospheric teleconnection in different month. The atmospheric teleconnection is derived from the analysis of winter blobs, which occur between November and March (Extended Data Table 1), as well as the preceding months. The authors' perspective is that this teleconnection is a result of stationary Rossby wave trains (Figures 2 and 4). In light of this viewpoint, it becomes important to carefully evaluate the impact of different background winds during various months on the atmospheric teleconnection. Background fields play a crucial role in the generation and guidance of Rossby waves, consequently influencing the establishment of distinct teleconnection patterns (e.g., Hoskins and Woollings 2015). Following the logic of examining prior months with lead times of 3, 2, 1, and zero, there is a concern that the teleconnection pattern observed

between November and February may differ significantly from the pattern observed between August and November. This disparity could also lead to inconsistencies in the mean state of the atmosphere, as utilized in the LBM simulations, which will be further discussed in the subsequent comment.

>>**Reply:**

Thanks so much for this crucial comment. Our previous composite logic appears not reasonable. We totally agree that background fields play a crucial role in the generation and guidance of Rossby waves, consequently influencing the establishment of distinct teleconnection patterns.

Accordingly, we re-plotted all the composite figures from September to following February based on the selected 13 warm blob winters. Results show that the Rossby wave activities and associated rainfall anomalies over the Mediterranean and North Atlantic regions are most significant in **November**. New numerical simulations (LBM and CAM5) are also performed based on real months. Thus, we mainly focus on results in **November** and present new figures based on the new composite logic in the revision. Please see details in our revised manuscript and supplementary information. We also revise the texts to present the analysis based on these new results.

The issue of causality in the study poses some challenges in interpretation. In Figure 1, the authors aimed to employ composite analysis with a lead month to provide insights into causality. However, two questions arise from this approach. Firstly, it is unclear why the strongest geopotential height anomalies do not occur at the peak time but rather at a lead of 1 month. Secondly, how can the authors be confident that there is not a third-party or confounding factor simultaneously driving both the blob and teleconnection phenomena? Lead-lag composite or correlation analyses alone may not be sufficient to reject such a possibility. Furthermore, Figure 2a presents composite meridional wind anomalies and WAF, but it remains questionable whether the WAF is causing the wind anomalies or vice versa, or by a confounding factor. We could also rationalize that the winds anomalies in the Mediterranean region are a response to upstream WAF emanating from the North Atlantic. The utilization of the Granger causality algorithm (e.g., Silva et al. in 2021) could potentially provide further insights in this regard, although I am not sure the authors have time or efforts to carry out the analysis in this study. The authors' inclusion of LBM simulations serves the purpose of addressing the causality issue. However, it is important to note that LBM simulations may also have their limitations, as described in the subsequent comment.

>>**Reply:**

Thanks for this critical comment.

1) The strongest geopotential height anomalies occur at 1 month leading the peak of the warm blobs, rather than coinciding with their peak, which is consistent with previous studies (e.g., Chen et al., 2021a,b). This result suggests that the response of SST is delayed due to the large heat capacity of sea waters to exhibit a noticeable warming. In the revision, we modified the composite method to illustrate the results in November (main texts) and October (supplement information).

2) In this paper, we **cannot** reject the possibility that a third-party factor is involved. We are convinced that the anomalous ridge over the NEP, **as part of the teleconnection**, is the direct driver of the warm blob by modulating the surface heat flux (Chen et al., 2021a,b). Even if a third-party factor exists, it still needs an **atmospheric bridge (i.e., atmospheric teleconnection)** to drive the warm blob. In the revision, we emphasize the combined role of anomalous cooling/heating over both the North Atlantic and the Mediterranean region to excite the wave trains/teleconnections and drive the anomalous ridge and the resultant warm blobs.

3) In Fig. 2a, we present WAF and meridional wind anomalies. The meridional wind anomalies are widely used to capture the horizontal wave-train pattern, while the WAF can display the propagation of Rossby wave energy. **We cannot conclude any causality in this figure, but WAF is crucial for the generation and maintenance of the wave train.** Hence, wind anomalies in the Mediterranean region are possibly a response to the upstream WAF emanating from the North Atlantic.

4) In the revision, we employ **both LBM and AGCM (i.e., CAM5)** to address the causality. Please see details in our revision.

5) We also tried to calculate the causality based on the Granger algorithm. First, we set the maximum lag time τ_{max} at 1 month. Then, we established the unconstrained regression equation $Hgt_{Nov} = \alpha_0 + \alpha_1 Hgt_{Oct} + \beta_1 Prep_{Oct} + \varepsilon_c(t)$. Hgt represents the average geopotential height over the NEP with subscripts for November (Nov) and October (Oct), respectively. $Prep$ represents average rainfall anomalies over the Mediterranean region. Climatology is calculated over 1991-2020. Anomaly is deviation relative to climatology. Then, we established the constrained regression equation $Hgt_{Nov} = \gamma_0 + \gamma_1 Hgt_{Oct} + \varepsilon_R(t)$. The F -statistic was constructed as:

$$F = \frac{(R_R - R_C)/\tau_{max}}{R_C/(N - 2\tau_{max} - 1)} = 6.85 > F_{0.1}(1,29) = 2.88$$

Thus, we rejected the original hypothesis and concluded that $Prep_{Oct}$ over the Mediterranean region is the Granger causal reason for the Hgt_{Nov} over the NEP.

However, considering the flow of this study, we do not include this part in the

revision.

3. The design of LBM simulations. In addition to the concerns regarding the background wind issue mentioned earlier, it is worth noting that the LBM used in this study appears to be a simplified representation of atmospheric dynamics. It is important to acknowledge that there is a significant disparity between the LBM and state-of-the-art global climate models (GCMs), such as CESM2, which are commonly utilized to investigate teleconnections and associated dynamics. Considering this disparity, it would be valuable for the authors to consider conducting GCM simulations to complement and support their findings based on observations and reanalysis data, as well as LBM simulations. By employing a GCM, they could potentially enhance the robustness of their conclusions by capturing a broader range of atmospheric processes and interactions. GCM simulations could provide a more comprehensive and realistic representation of the teleconnection phenomenon under investigation. To further address the causality issue, it could be beneficial to conduct a series of LBM simulations (and potentially GCM simulations) with forcings specifically applied to hotspots in southeast China, the Ural Mountain region, or the North Atlantic. By doing so, the authors can examine whether a similar teleconnection pattern emerges, as indicated by the statistically significant RWS presented in Figure 2b. These targeted simulations would provide an opportunity to assess the robustness of the observed teleconnection pattern and its relationship to the specified forcings. If inconsistent teleconnection patterns are observed across multiple simulations with different forcings applied to these specific regions, it would lend support to the hypothesis that the Mediterranean regions play a causal role in driving the teleconnections. This approach would strengthen the authors' argument regarding the causality between the identified hotspots and the observed teleconnection pattern.

>>Reply:

We really appreciate this helpful comment.

1) In the revision, we have modified the background condition of the LBM experiments by fixing it to November. We also acknowledge the disparity between the LBM and state-of-the-art global climate models (GCMs). According to your suggestion, we conduct several additional numerical experiments based on CAM5 by imposing prescribed SST forcing. By incorporating a broader range of atmospheric processes and interactions, the anomalous ridge over the NEP as well as the related extratropical wave trains are also reasonably captured (Fig. 5 and Extended Data Fig. 10), which further support our findings based on observations/reanalyses.

2) In the revision, we conducted two types of numerical experiments based on the

LBM and CAM5. For the LBM, 4 major forcings over the North Atlantic, Mediterranean region, southern China and surrounding seas, and northern China as well as surrounding seas, were imposed based on the RWS and anomalous heating distribution. For CAM5, we mainly imposed SST forcings over the North Atlantic and Mediterranean regions. Based on both observations and simulations, we highlight the combined role of the North Atlantic and Mediterranean regions in driving the extratropical wave trains along the jet streams and anomalous ridge over the NEP.

Minor comments:

Line 46: What is salinity variation? Can the authors discuss more about its impact on blob?

>>Reply:

Here we are describing the mixed layer depth changes related to the salinity anomaly. Zhi et al. (2019) documented that the early negative salinity anomaly dominantly contributed to the shallower mixed layer depth (MLD) in the NEP during the period of 2012–2013. Then, the shallower mixed layer trapped more heat in the upper water column and resulted in a warmer SST, which enhanced the warm blob. The salinity variability contributed to approximately 60% of the shallowing MLD related to the warm blob.

We modified this sentence to make it clear (RL 48).

Line 65: Are all blobs peaking in boreal winter? I saw in the Extended Data Table 1 some blobs peak in March and November, which may not be typical boreal winter months: December, January, and February. How about using cold season in the Northern Hemisphere?

>>Reply:

Thanks for this comment. We have modified related places into “cold season” or “boreal cold season” to make it more accurate.

Line 74: do you use TNH abbreviation in the following part of manuscript?

>>Reply:

We did not use TNH abbreviation in the following part of our manuscript. Thus, we no longer include the TNH abbreviation in the revised manuscript.

Lines 76-77: do the authors mean wintertime midlatitude wave trains? Or autumn wave trains?

>>Reply:

Thanks for this careful suggestion.

In the revision, we only focused on **autumn** wave trains, especially in November (Fig. 2a) and October (Extended Data Fig. 3a).

Lines 76-83 and 84-86: Why the wave train, originating in the North Atlantic, does not coincide with the WAF, from Mediterranean region?

>>Reply:

The wave trains originating from the North Atlantic have two major pathways: along subtropical and subpolar jet streams. The subtropical wave train is enhanced over the Mediterranean region. Thus, the wave train coincides well with the WAF over the Mediterranean region.

We modified related sentences in RL 87–98.

Lines 99-100: A recent study (Hong and Hsu 2023) showed that the central Pacific could affect Northeast Pacific. How do the authors reconcile their results and those presented in this study?

>>Reply:

Apologies that our text was misleading and thank you for identifying this paper. We did not mean the tropical Pacific cannot affect SST in the Northeast Pacific. We agree that the tropical Pacific can exert a huge influence on the atmospheric circulation and SST over the Northeast Pacific. But our paper just focuses on the additional important contribution from the subtropical and subpolar Rossby wave trains as a teleconnection. In fact, the effects from the tropical central Pacific cannot be well reflected from the WAF due to the limitation of its calculation.

To prevent further misleading, we deleted this sentence in the revision.

Lines 108-110: why the wave trains often bifurcate into two branches over the Mediterranean? Can the authors discuss more?

>>Reply:

The reasons for this wave train bifurcation are: 1) the wave train near the Mediterranean region is located outside of the area with a strong absolute vorticity gradient such that it is partially reflected, and 2) it is sensitive to the origin of the Rossby wave.

We add more discussion in the revision and cite the related two papers (RL 118–121).

Lines 136-137: can the authors show their evolution?

>>Reply:

We show the composite rainfall anomalies in November (Fig. 3a) and October

(Extended Data Fig. 5) in the revision.

Lines 154-155: Does the strongest magnitude happen at 300 hPa because of the vertical resolution? If the vertical resolution increases, could the magnitude shift to, for example, 309 hPa?

>>Reply:

The specific level of strongest heating can change with the vertical resolution. But both the observation and model results do not have such fine resolution into 309 hPa. We use 12 vertical levels to show the vertical profiles.

Moreover, we have verified that the response of the geopotential height is not very sensitive to the specific vertical level with maximum heating (e.g., Figs. 4a–c vs. Extended Data Fig. 8).

Lines 158-159: what are the background mean fields given in the LBM simulation? Can the authors show their spatial maps?

>>Reply:

We show the spatial maps of the background mean fields given in the LBM simulations (Fig. R2 below). We also add this figure to the supplementary information (Extended Data Fig. 6).

Fig. R2 Background climatology of (a) geopotential height (shading, in gpm) and zonal wind (contour, in m/s) at 300 hPa, (b) geopotential height (shading, in gpm) at 500 hPa, and (c) SLP (shading, in hPa) and surface winds (vector, in m/s) in November given in LBM.

Line 231: As the authors had calculated three-dimension WAF, I am curious what is the role of vertical component here?

>>Reply:

Thanks for this comment.

Equation (1) in original Line 231 is actually the horizontal WAF, not the three-dimensional WAF. The vertical component is too small, which is not used very often. But we plot the figure including the vertical and zonal WAF averaged over 35–45°N (Fig. R3 below). After multiplying 2000 for the vertical component, we can see downward wave energy into the NEP, which may play a role for the anomalous ridge there. There is also upward wave energy in the middle-upper troposphere of the

Mediterranean region, but it is very small.

Considering the small magnitude of the vertical component of the WAF, we did not add this figure to the manuscript.

Fig. R3 As in Fig. 2a, but for the vertical-zonal WAF distribution. Note that the vertical component is multiplied by 2000. Red and green lines outline the Mediterranean and NEP regions, respectively.

Line 268: what does the gamma-type mean?

>>**Reply:**

There are 3 types of vertical profile of forcing: 1) sinusoidal, 2) gamma, and 3) uniform. The gamma-type profile has one central level with the maximum forcing, which is related to the gamma function. This type of vertical profile is closer to the real heating profile.

In the revised manuscript, we provide the figures of vertical profiles in the LBM and have deleted the “gamma-type” description to prevent potential misunderstanding.

Line 455, Figure 1L since the authors mentioning TNH pattern, which has anomaly centers in the North Atlantic, can you extend the domain to the North Atlantic?

>>Reply:

We extend the domain to the North Atlantic to show the circulation anomalies in the revision. Please see revised Fig. 1 and Extended Data Fig. 2 for details.

Extended Data Table 1: the bottom two events are the same, except “2019” and “2020”. Can the authors clarify?

>>Reply:

Thanks for this careful comment.

According to Chen et al. (2021a), the warm blob exists continuously from May 2019 to December 2020. It has two peaks, respectively in November of 2019 and 2020. But considering that two winters (or cold seasons) are involved in this continuous event, we just separate it into two events shown in the Extended Data Table 1 and our further analysis.

References:

- Chen, Z., Shi, J., Liu, Q., Chen, H. & Li, C. A persistent and intense marine heatwave in the Northeast Pacific during 2019–2020. *Geophys. Res. Lett.* 48, e2021GL093239 (2021a).
- Chen, Z., Shi, J. & Li, C. Two types of warm blobs in the Northeast Pacific and their potential effect on the El Niño. *Int. J. Climatol.* 41, 2810–2827 (2021b).
- Shi, J., Tang, C., Liu, Q., Zhang, Y., Yang, H. & Li, C. Role of mixed layer depth in the location and development of the Northeast Pacific warm blobs. *Geophys. Res. Lett.* 49, e2022GL098849 (2022).
- Tang, C., Shi, J. & Li, C. Long-lived cold blobs in the Northeast Pacific linked with the tropical La Niña. *Clim. Dyn.* 57, 223–237 (2021).
- Zhi, H., Lin, P., Zhang, R.-H., Chai, F. & Liu, H. Salinity effects on the 2014 warm “Blob” in the Northeast Pacific. *Acta. Oceanol. Sin.* 38, 24–34 (2019).

Reviewer #2:

This paper looks at the potential role of the Mediterranean region in forcing “warm blobs” in the northeast Pacific by inducing an anomalous atmospheric ridge in that region. The authors claim that enhanced rainfall over the Mediterranean drives a wave train in the northern hemisphere that generates this anomalous ridge, which subsequently leads to marine heatwave events in the North Pacific. The manuscript is well written, but I am unconvinced regarding the causality based on the evidence that has been presented. My main comments are as follows:

>>Reply:

Thanks for your very valuable comments that have helped us further improve the manuscript. Our revised manuscript should now hopefully provide a more compelling demonstration of this mechanism. Throughout our responses below, for convenience, the line numbers in the new revised manuscript are referred to as Revision Lines (RL).

Previous studies have linked Northeast Pacific marine heatwaves to the tropics. While the authors reference the link to ENSO and the tropics in the introduction, as far I can tell it is not accounted for in this study nor has the influence of ENSO or the tropics been removed in any way. Therefore I don't think it's possible to rule out the role of the tropics in forcing these extratropical wave trains or at least contributing towards the marine heatwave/anomalous North Pacific anticyclone. Related to this, the rainfall anomaly plots (Fig 3b and extended data fig 4) show a very limited area. It would be useful to see the rainfall anomalies including the rest of the northern hemisphere (including the tropics), to see any other regions of enhanced or suppressed rainfall, as I find it hard to believe that the tropics is not forcing the extratropics in any way. Many areas within the Mediterranean region don't actually have significant rainfall anomalies in most months prior, either, even with the fairly generous significance threshold used (extended data fig 4)

>>Reply:

Thanks for providing this critical comment.

We do indeed agree that tropical processes (e.g., ENSO, rainfall anomalies etc.) are crucial for exciting wave trains to propagate into mid-latitude regions (e.g., NEP).

There have been a large number of papers that demonstrate and discuss these issues.

However, our study only focuses on the mid-latitude Rossby wave sources (i.e., mainly North Atlantic and Mediterranean region contributions) to trigger/maintain the anomalous ridge over the NEP. Although these sources can be induced by tropical processes, they are beyond the scope of this study. As you

raised above, composite analysis cannot rule out the potential role of the tropics in forcing the extratropical wave trains and anomalous ridge over the NEP. **Here, we conducted numerical simulations using an atmospheric linear baroclinic model (LBM) and AGCM (CAM5) to isolate the contributions from the North Atlantic and Mediterranean regions.**

In terms of rainfall anomalies, we show rainfall anomalies **including the tropical area** in November (Fig. R4 below). There are regions with enhanced or suppressed rainfall in the tropics, which can excite teleconnections. In addition, we modified the composite method to illustrate the results in **November** in the revision. The rainfall anomalies over the North Atlantic and Mediterranean regions are much stronger and are statistically significant in our new results.

Fig. R4 Composite rainfall anomalies (shading, in mm) in November of warm blobs peaking in winter. Stippling indicates exceeding 0.1 significance level based on a two-tailed Student's *t*-test. Black box marks the Mediterranean region.

To reveal the potential role of tropical rainfall anomalies on extratropical atmospheric circulation anomalies in **November**, we conducted a series of experiments using the LBM. When anomalous cooling is added over the Maritime Continent, geopotential height anomalies are not robust in the extratropical regions (Fig. R5). Geopotential height anomalies are also not evident over the extratropics when anomalous heating is prescribed over the tropical Indian Ocean (Fig. R6). But when anomalous heating is imposed over the tropical western Pacific, the extratropical response is prominent with the anomalous ridge shifting a bit northward and overwhelmed by an anomalous trough over the NEP (Fig. R7). Note that the atmospheric circulation response is sensitive to the background climatology. If similar cooling is prescribed over the Maritime Continent with a February background, the geopotential height response is much stronger (Fig. R8). **Nevertheless, we do not focus on the extratropical teleconnection excited directly by tropical processes. Thus, our above results are not included in**

our manuscript.

Fig. R5 Same as Extended Data Fig. 7, but imposing anomalous cooling over the Maritime Continent.

Fig. R6 Same as Extended Data Fig. 7, but imposing anomalous heating over the tropical Indian Ocean.

Fig. R7 Same as Extended Data Fig. 7, but imposing anomalous heating over the tropical western Pacific.

Fig. R8 Same as Extended Data Fig. 7, but imposing anomalous cooling over the Maritime Continent with a February background.

While wave activity flux can be a useful diagnostic, it doesn't give any information about causality. The authors state that "Anomalous Rossby wave energy, indicated by wave activity flux, flows from the Mediterranean region into the subtropical jet". While there are wave activity flux vectors over the Mediterranean, there are also vectors in most other parts of the extratropics and there is no evidence here to suggest that the waves originate over the Mediterranean itself. The vectors are also scaled in such a way on the plot that they are difficult to distinguish. I also don't think that wave activity flux is particularly good at showing propagation of waves from the tropics to extratropics, which I think is the only evidence the authors use to rule out the role of the tropics. To me, all this panel shows is waves propagating around the northern hemisphere but with no clear information about where they originate or what is forcing them. This is particularly the case when looking at the monthly figures (extended data fig 2)

>>Reply:

1) In our revised manuscript, we modified the above-mentioned sentence into "Rossby wave energy, indicated by wave activity flux (WAF; Methods), greatly intensifies over the Mediterranean region (vectors in Fig. 2a), implying a potential Rossby wave source (RWS) in this region". Hence, we show the origin of Rossby waves in terms of RWS, rather than WAF.

2) We re-plotted the figures related to the WAF in the revision (Fig. 2a and Extended Data Fig. 3a) to make it clear.

3) We agree that the WAF cannot reasonably show propagation of waves from the tropics. But we did not use the WAF to rule out the role of the tropics. We have deleted/modified the related sentences to prevent potential misleading.

4) We agree that the WAF cannot show any causality. The WAF can capture wave propagation features. The origin of Rossby waves is calculated in terms of the RWS variable. Moreover, we verify our causality hypotheses by performing numerical simulations (LBM and CAM5).

The Rossby wave source plot (Fig 2b) has many areas of significant source, so why the focus on the small area over the Mediterranean? Many of these areas of RWS will also be generated from tropical rainfall, which doesn't seem to have been taken into account.

>>Reply:

In the revision, we focus on **not only on the Mediterranean region but also the North Atlantic**. The reasons for these choices are mainly: 1) They have large areas of significant RWS; 2) There are prominent WAF or WAF intensification over these two areas. For the composite results in November in the revision (Fig. 2b), significant RWS areas over the Mediterranean and North Atlantic regions are **not small** compared to other areas. Moreover, we also discuss the two areas over China and surrounding seas.

We agree that some areas with significant RWS can be generated from tropical rainfall. But their drivers are beyond the scope of this study. Our study only aims to identify the extratropical source regions of Rossby waves that contribute to the anomalous ridge over the NEP, which is responsible for the generation of the warm blobs.

It seems to me that what is more likely is that the anomalous ridge over the North Pacific is actually influencing Mediterranean rainfall through perturbing the North Atlantic jet. Again, this doesn't seem to have been considered in the present study. This is also made harder to see by the plots being cut off between 90W-0 longitude. Related to this, there seems to be a break in the wave train over central Asia (Fig 2), which would also support the North Pacific influencing Europe, rather than the other way round.

>>Reply:

Thanks for drawing attention to this alternative possibility.

To test this, we have conducted an additional experiment **by imposing anomalous SST warming over the NEP** in CAM5 (Fig. R9). The detailed design is similar to that of Exp_Med_ideal (see Methods and Extended Data Table 2) but with the forcing location over the NEP (i.e., green box in Fig. 1). When the NEP is warming, enhanced

rainfall is only identified over the Mediterranean region in **December (Fig. R9d)** and **February (Fig. R9f)**, while near normal or drier conditions are found over the Mediterranean region in other months (Figs. R9a,b,c,e). Thus, it is much more likely that the Mediterranean rainfall-induced teleconnection influences the atmospheric circulation over the NEP guided by jet streams before December. The blob-like warming over the NEP can exert effects on the enhanced Mediterranean rainfall, but from December.

In addition, the WAF is continuous over Eurasia when compositing the warm blob events relative to natural month (i.e., November shown in this paper) in the revision.

Fig. R9 Response of rainfall (shading, in mm) to prescribed warm SST forcing over the NEP in CAM5.

The authors state that the LBM experiment “closely reproduces the anomalous ridge over the NEP”, which seems like a bit of a strong statement to me. Yes, there is an anomalous ridge over the North Pacific but it is shifted south somewhat. The wave activity flux is also extremely messy which makes it hard to see what’s going on, but this doesn’t seem to correspond very closely to Fig 2a either. I suspect that if a similar heating was applied elsewhere in the northern hemisphere (e.g. north of India) that a similar pattern would emerge, so I don’t think this provides sufficient evidence of the Mediterranean forcing the North Pacific.

>>Reply:

Thanks for this crucial comment.

1) We agree that the LBM cannot simulate the exact location of the anomalous ridge over the NEP. There is always a shift in its location between modeled and observational results. The anomalous ridge, higher-than-normal sea level pressure, and associated easterly anomalies over the NEP are in good accordance with those in observations.

2) The pathway of the WAF based on our new LBM experiments is clearer when fixing the background to November. In addition, we did not conduct LBM experiment/s by imposing heat over north of India because the anomalous heating there is not robust in November (Fig. 3b).

3) To provide more evidence to the above connection, we also conducted CAM5 simulations by imposing anomalous SST over the Mediterranean and North Atlantic regions.

Reviewer #3:

The paper describes an interesting possible connection between anomalous heating in the Mediterranean region and North Pacific “warm blobs”. The results are potentially new, but they need to be better substantiated and supported by more analysis. The paper is well structured, but a few points need to be clarified before being considered for publication.

>>Reply:

Thanks for your positive and valuable comments that have helped us further improve the manuscript. Throughout our responses below, for convenience, the line numbers in the new revised manuscript are referred to as Revision Lines (RL).

1. The discussion in lines 76–83 need a better motivation. The wavetrain is identified only by two contours and the amplitude over the Mediterranean is minor. The WAF seems to be distributed over a much larger pattern over Europe rather than only over the Med, the RWS has a smaller spatial scale by construction and exhibit many other maxima, for instance over North Europe. In summary, it is hard to conclude that the Med plays any special role.

>>Reply:

Thanks for this valuable comment.

- 1) The two contours of zonal winds depict the climatological location of jet streams (Fig. 2a). We did not identify wave trains by this variable.
- 2) In the revision, we highlight two wave trains: one intensifying from the Mediterranean region and propagating along the subtropical jet stream, and another propagating from the North Atlantic through Europe along the subpolar jet stream. We focus on the Mediterranean region because there is a robust WAF intensification in this region. Moreover, meridional wind anomalies, indicative of the wave train pattern, are significant over the Mediterranean region.
- 3) In terms of the RWS, our revised composite results based on November show a wider area with significant RWS over the Mediterranean region. Three other areas (North Atlantic and China and surrounding seas) are also discussed in the revised manuscript. By contrast, although there may be other significant RWS places, they have much smaller spatial scale. We do not present specific discussion on them.

2. The vertical structure of the heating computed as a residual is surprising (Fig, 7 Extended data and other). At these latitudes is mostly shallow heating (see for instance Ling and Change, 2013), so the profile showing maximum heating at 300mb is rather surprising.

>>Reply:

Thanks for this careful comment. We carefully read this paper.

We agree that it is mostly shallow heating over our focused area. We also re-plotted the related vertical profiles in Ling and Zhang (2013).

However, the LBM is **linearized** about a basic state. We imposed an **anomalous** heating vertical profile to the model. Figures 4a and 4d show the **anomalous** profiles of heating, rather than their original values, over the Mediterranean and North Atlantic regions, respectively. We have double-checked these profiles. In the Mediterranean region, anomalous heating does not change largely from 700 hPa to 300 hPa.

We modified the captions of the related figures and emphasize the Q1 “**anomaly**” to make it clear.

3. The results from the linear model also need to be better justified. The model needs to be described better, especially regarding the dissipation formulation used and the vertical structure of the heating imposed. The cryptic statement at the end of the method section (“A gamma-type ...”) needs to be better formulated.

>>Reply:

Thanks for this careful comment.

First, we now also present results based on a state-of-the-art AGCM (i.e., CAM5) in the revision to justify those based on the LBM.

Second, we provide greater description of the LBM in the revised manuscript:

1) In terms of dissipation, the LBM includes three dissipation terms: a bi-harmonic horizontal diffusion with the damping timescale of 1 day for the smallest wave, very weak vertical diffusion (damping timescale of 1000 days) to remove a vertical noise arising from finite difference, and the Newtonian damping and Rayleigh friction as represented by a linear drag, which has a timescale of 1 day applied only to the lower boundary layers and the uppermost two levels (Watanabe and Jin, 2003).

2) There are 3 types of vertical profile of forcing: 1) sinusoidal, 2) gamma, and 3) uniform. The gamma-type profile has one central level with the maximum forcing, which is related to the gamma function. This type of vertical profile is closer to the real heating profile.

In the revised manuscript, we added the above sentences related to the dissipation in RL 301–306. We provide the figures of the vertical profiles in the LBM and deleted the “gamma-type” description to prevent potential misunderstanding.

4. Assuming that the solution has converged successfully, the way it develops is suspect. The development of the response in the Pacific can be due to residual amplitude along

the equatorial region deriving from the spectral truncation of the localized anomaly prescribed over the Mediterranean. At T42 resolution the expansion of the localized heating over the spectral transform will generate amplitude along the equator that will be picked up by the model in the transform cycle and result in further forcing for the mid-latitudes. This is particularly important in the West Pacific because the basic state is very sensitive to forcing there and it can amplify easily. Depending on the dissipation and cooling imposed this effect may be more or less emphasized, so the modeling choices are very important to clarify.

>>Reply:

Thanks for providing this important comment.

- 1) According to your comment above, we agree that there may be some residual heating along the equator due to expansion of the localized heating in the transform cycle. Nevertheless, we can see clear propagation of Rossby waves along jet streams in terms of the WAF (Fig. 4). The geopotential height response is also confined with extratropical regions. We think the effects from extratropical regions (e.g., Mediterranean and North Atlantic regions) are well revealed from the above two aspects.
- 2) We have clarified the details in terms of dissipation in the revised Methods.
- 3) In addition, we also performed CAM5 simulations to further verify our results. We used f19_g16 resolution (i.e., $2.5^{\circ} \times 1.9^{\circ}$) for the experiments. There are no SST anomalies outside of the forcing area. In other words, SST was prescribed to climatology outside of the forcing area.

References:

- Watanabe, M., & Jin, F.- F. A moist linear baroclinic model: Coupled dynamical–convective response to El Niño. *J. Clim.* 16(8), 1121–1139 (2003).
- Ling, J., & Zhang, C. Diabatic heating profiles in recent global reanalyses. *J. Clim.* 26(10), 3307–3325 (2013).

REVIEWER COMMENTS

Reviewer #1 (Remarks to the Author):

I appreciate the authors' dedication to addressing my comments and enhancing the manuscript's quality. Many of my concerns have been appropriately handled. However, I have a few additional remarks for the authors to consider.

The removal of linear trends. The authors have successfully addressed my previous concerns. However, I would like the authors to consider the non-linearity in the SST response even after removing the linear trend, as discussed in several prior studies (Trenberth and Shea 2006; Zhang et al. 1997; Mantua et al. 1997; and others). It might be beneficial to explore the impact of quadratic trends removal, although I understand that this may introduce additional complexities and the results may not be varied much. I also suggest the authors to review more relevant papers to rationalize the removal of linear trend.

November atmospheric circulation. While I appreciate the shift in focus to atmospheric circulation in November for understanding the underlying mechanism, I have a concern upon rereviewing Extended Data Table 1. It appears that a significant number of blob events did not occur in November. Consequently, I am curious about the applicability of the November-based mechanism to other months when blob events are observed. Clarification on this point would enhance the robustness of the proposed mechanism.

References

- Mantua, N. J., S. R. Hare, Y. Zhang, J. M. Wallace, and R. C. Francis, 1997: A Pacific interdecadal climate oscillation with impacts on salmon production. *Bull. Amer. Meteor. Soc.*, 78, 1069–1079, doi:10.1175/1520-0477(1997)0782.0.CO;2.
- Trenberth, K., and D. Shea, 2006: Atlantic hurricanes and natural variability in 2005. *Geophys. Res. Lett.*, 33, L12704, doi:10.1029/2006GL026894.
- Zhang, Y., J. M. Wallace, and D. S. Battisti, 1997: ENSO-like interdecadal variability: 1900–93. *J. Climate*, 10, 1004–1020, doi:10.1175/1520-0442(1997)0102.0.CO;2

Reviewer #2 (Remarks to the Author):

I appreciate the authors' efforts to address my concerns. I still have some further comments that should be addressed, however.

Firstly, with regards to the influence of tropical forcing on the development of the warm blobs, I

understand that the authors are focusing on the extratropical teleconnections, but some mention of the possible role of the tropics should be made. While the LBM and CAM5 experiments demonstrate that heating/cooling in the North Atlantic/Mediterranean can generate anomalies over the North Pacific, I still don't think that the anomalies in these experiments are particularly close to those shown in Figure 1. For example, the positive height anomalies in Extended Data Figure 10 are shifted quite a long way north and east. Therefore I wonder if a combination of forcing from the tropics and the Mediterranean is needed in order to recover anomalies over the North Pacific that are more similar to those that are observed. I'm not sure if there is a way to test this in the current experimental setup, but I think that at least mentioning the contribution of the tropics in the text would be helpful as currently the text reads as if it is suggesting that the Mediterranean is the dominant driver of the warm blobs, which I don't think is justified.

Some of the figures are still hard to interpret, particularly those which have WAF flux on them. In particular, the arrows on Figures 2a and 4 are extremely hard to distinguish (Figure 4 is especially messy). Improving this would aid interpretation of the results

The paragraph beginning on Line 222 seems out of place. Currently the manuscript describes LBM experiments, then CAM5 experiments and back to LBM. It would make sense to include this paragraph with the description of the other LBM experiments

Minor comments/typos:

It would probably be good to be consistent with the labelling of subplots on figures. On some figures the panels a, b, c, d etc are in order going down each column, and on some they go left to right.

Line 79: Typo in "eastern"

Line 85: Typo in "intensification"

Reviewer #3 (Remarks to the Author):

I appreciate the work that you have put in the paper, but I think that are substantial issues still left open.

1. The model description is still lacking, it is good to have now a description of the dissipation, but still is not described how the solution have been obtained, nor how the convergence has been monitored. Confirming results with a direct method would have been more convincing.

2. The structure of the heating is still puzzling. It is also unclear how the vertical structure of the heating has been derived. Even considering the imposed heating (red line in Fig.4) it is concentrated at 200mb, implying really high level anomalous convection that is not typical of the Mediterranean area. In any case, the balances at this latitudes require that heating is compensated by meridional motion, not vertical motion, negating a Gill-like response that would generate a high level vorticity source. So it is likely that the response that you see is generated by a quasi-resonant response of the linear model, due to ill-conditioning of the linear system. Evidence of this can be seen in the almost identical response in

panels (b) and (e) except for the sign that of course is a consequence of the forcing having two different signs. By the way, it is also strange that, given the similarity in the 500mb Geopotential the SLP is different.

The nonlinear experiments in Fig. 5 do not help much. First of all, the experiments are not described (which climatology is subtracted ? how they have been initialized ?), then they show that basically the perturbations tend to generate instabilities in the most sensitive areas of the jets, over the Pacific/North American sector and over the Atlantic. Also in this case, it is hard to judge that the Med and Atl response are significantly different, for instance, from putting perturbations in any other area.

In summary, I think that the point of the remote influence of the Med on the North Pacific has not been sufficiently demonstrated.

REVIEWER COMMENTS

Reviewer #1 (Remarks to the Author):

I appreciate the authors' dedication to addressing my comments and enhancing the manuscript's quality. Many of my concerns have been appropriately handled. However, I have a few additional remarks for the authors to consider.

>>Reply:

We very much appreciate your constructive comments that have helped us further improve the manuscript. Throughout our responses below, for convenience, the line numbers in the new revised manuscript are referred to as Revision Lines (RL).

1. The removal of linear trends. The authors have successfully addressed my previous concerns. However, I would like the authors to consider the non-linearity in the SST response even after removing the linear trend, as discussed in several prior studies (Trenberth and Shea 2006; Zhang et al. 1997; Mantua et al. 1997; and others). It might be beneficial to explore the impact of quadratic trends removal, although I understand that this may introduce additional complexities and the results may not be varied much. I also suggest the authors to review more relevant papers to rationalize the removal of linear trend.

>>Reply:

Thank you for raising this and suggesting several related papers. Here, we show the quadratic trends of SST within the study area (Fig. R1). After removing these trends, the blob indices were highly consistent with a correlation coefficient of 0.97 (Fig. R2). However, the selected warm blob events were not totally consistent because of our criterion in intensity and duration (Table R1). For example, there were no warm blob events in the period 2019–2021 based on quadratic detrending because the blob index was somewhat smaller compared to that using linear detrending. Thus, although the blob index was also positive and greater than 1.0 for many months, they cannot persist for more than five months (not shown). We further show the same figures in the main text (i.e., Figs. 1–3) based on the warm blob events selected by quadratic-detrended blob index (Figs. R3–R5). These results are very similar with our original results.

Nevertheless, we did not change the warm blob events in the revised manuscript to keep consistency with previous studies (e.g., Chen et al., 2021a,b). Moreover, the warm blob events in the period 2019–2021 have been widely recognized in recent studies (Amaya et al., 2020; Chen et al., 2021b; Scannell et al., 2020), which should be discussed in this study.

Although we did not change the warm blob events, we have added further discussion in RL 262–264 of our revised manuscript.

Fig. R1 Normalized time series (black line) of SST anomalies averaged over the study area of the warm blob. Blue and red lines represent the linear and quadratic trends, respectively.

Fig. R2 Blob indices with linear (blue line) and quadratic (red line) removal of long-term trends.

Table R1: List of NEP warm blob events with their start time and end time. Events with bold fonts indicate those that are different identified by two warm blob indices.

Linear detrend	Quadratic detrend
Dec 1956 – Jun 1957	Dec 1956 – May 1957
Oct 1961 – Sep 1962	Oct 1961 – Sep 1962
Jan 1963 – Sep 1963	Jan 1963 – Sep 1963
Mar 1965 – Sep 1965	Mar 1965 – Sep 1965
Oct 1985 – Mar 1986	Dec 1978 – Jul 1979
Sep 1989 – Jan 1990	Sep 1985 – Mar 1986
Jan 1991 – Jun 1991	Aug 1989 – Feb 1990
Oct 1993 – Feb 1994	Nov 1990 – Jun 1991
Nov 2013 – Jun 2014	Oct 1991 – Jun 1992
Feb 2015 – Sep 2015	Sep 1993 – Mar 1994
Jun 2019 – Dec 2019	Feb 1997 – Jun 1997
Apr 2020 – Dec 2020	Dec 2004 – Apr 2005
May 2021 – Sep 2021	Nov 2013 – Jun 2014
	Feb 2015 – Aug 2015

Fig. R3 Same as Fig. 1, but for warm blob events with quadratic removal of SST trends.

Fig. R4 Same as Fig. 2, but for warm blob events with quadratic removal of SST trends.

Fig. R5 Same as Fig. 3, but for warm blob events with quadratic removal of SST trends.

trends.

2. November atmospheric circulation. While I appreciate the shift in focus to atmospheric circulation in November for understanding the underlying mechanism, I have a concern upon re-reviewing Extended Data Table 1. It appears that a significant number of blob events did not occur in November. Consequently, I am curious about the applicability of the November-based mechanism to other months when blob events are observed. Clarification on this point would enhance the robustness of the proposed mechanism.

>>Reply:

As shown in Extended Data Table 1, four out of 13 warm blob events did not occur in November. We note that anomalous warming was evident for these four cases over the study area (Fig. R6), although their intensity was not as strong as 1.0 standard deviation. The exception was the case in 1962 (Fig. R6b), with anomalous warming occurring in December 1962.

The mechanisms proposed in the present study **may not be applicable** for other boreal winter months (Fig. R7) because background fields play a crucial role in the generation and guidance of Rossby waves, and influence the establishment of teleconnection patterns. In fact, as the circulation anomalies in terms of geopotential height anomalies at 500 hPa were most significant in November (Fig. 1), the wave train mechanisms were crucial for the anomalous ridge and the warm blobs in the study area.

We clarify this point in RL 237–240 regarding the robustness and validity of our proposed mechanisms.

Fig. R6 SST anomalies in November of (a) 1956, (b) 1962, (c) 1990, and (d) 2014.

Fig. R7 Same as Fig. 2b, but in (a) September, (b) October, (c) November, (d) December, (e) January, and (f) February.

References:

- Amaya, D. J., Miller, A. J., Xie, S. P., & Kosaka, Y. Physical drivers of the summer 2019 North Pacific marine heatwave. *Nature Commu.* 11, 1903 (2020).
- Chen, Z., Shi, J., Liu, Q., Chen, H. & Li, C. A persistent and intense marine heatwave in the Northeast Pacific during 2019–2020. *Geophys. Res. Lett.* 48, e2021GL093239 (2021a).
- Chen, Z., Shi, J. & Li, C. Two types of warm blobs in the Northeast Pacific and their potential effect on the El Niño. *Int. J. Climatol.* 41, 2810–2827 (2021b).
- Scannell, H. A., Johnson, G. C., Thompson, L., Lyman, J. M., & Riser, S. C. Subsurface evolution and persistence of marine heatwaves in the Northeast Pacific. *Geophys. Res. Lett.* 47, e2020GL090548 (2020).

Reviewer #2 (Remarks to the Author):

I appreciate the authors' efforts to address my concerns. I still have some further comments that should be addressed, however.

>>Reply:

Thank you for your very valuable comments that have helped us further improve the manuscript. Throughout our responses below, for convenience, the line numbers in the new revised manuscript are referred to as Revision Lines (RL).

Firstly, with regards to the influence of tropical forcing on the development of the warm blobs, I understand that the authors are focusing on the extratropical teleconnections, but some mention of the possible role of the tropics should be made. While the LBM and CAM5 experiments demonstrate that heating/cooling in the North Atlantic/Mediterranean can generate anomalies over the North Pacific, I still don't think that the anomalies in these experiments are particularly close to those shown in Figure 1. For example, the positive height anomalies in Extended Data Figure 10 are shifted quite a long way north and east. Therefore I wonder if a combination of forcing from the tropics and the Mediterranean is needed in order to recover anomalies over the North Pacific that are more similar to those that are observed. I'm not sure if there is a way to test this in the current experimental setup, but I think that at least mentioning the contribution of the tropics in the text would be helpful as currently the text reads as if it is suggesting that the Mediterranean is the dominant driver of the warm blobs, which I don't think is justified.

>>Reply:

Thank you for raising this very important point.

Yes, we agree that the geopotential height anomalies in the LBM and CAM5 experiments are shifted to some extent, implying the potential role from the tropics or other regions.

First, to test the potential role of the tropical oceans, we imposed anomalous heating/cooling in November in the LBM, as we responded last time (Figs. R8–R10). When anomalous cooling was added over the Maritime Continent (Fig. R8) and tropical Indian Ocean (Fig. R9), geopotential height anomalies are not robust in the extratropical regions. But when anomalous heating was imposed over the tropical western Pacific, the extratropical response is prominent with the anomalous ridge shifting a bit northward (Fig. R10).

Moreover, we also performed three additional experiments with forcings imposed in the tropical Pacific (EXP_TP-only), tropical Pacific and North Atlantic (EXP_TP+NAtl), and tropical Pacific and Mediterranean (EXP_TP+Med) regions

using CAM5 (Fig. R11). The detailed model set-up is similar to that explained in the main text. By comparing with Fig. 5, the combined effects of the tropical Pacific and Mediterranean region contributions produce a more realistic location of the anomalous ridge over the Northeast Pacific (Fig. R11c). However, for the TP-only run, the patterns of geopotential height anomalies are similar to those in Fig. 5 with a slightly stronger intensity (Figs. R11a and R11d).

We have now added the important contribution of the tropics in the combined experiments which now provide more compelling and representative results (RL 240–242; Extended Data Fig. 13).

Fig. R8 Same as Extended Data Fig. 7, but imposing anomalous cooling over the Maritime Continent.

Fig. R9 Same as Extended Data Fig. 7, but imposing anomalous heating over the tropical Indian Ocean.

Fig. R10 Same as Extended Data Fig. 7, but imposing anomalous heating over the tropical western Pacific.

Fig. R11 (a,b,c) Same as Fig. 5b, but for EXP_TP-only run, EXP_TP+NAtl run, and EXP_TP+Med run, respectively. (d,e,f) Same as (a,b,c), but for November.

Some of the figures are still hard to interpret, particularly those which have WAF flux on them. In particular, the arrows on Figures 2a and 4 are extremely hard to distinguish (Figure 4 is especially messy). Improving this would aid interpretation of the results

>>Reply:

We have re-plotted Figures 2 and 4 to make them clearer in our revised manuscript.

The paragraph beginning on Line 222 seems out of place. Currently the manuscript describes LBM experiments, then CAM5 experiments and back to LBM. It would make

sense to include this paragraph with the description of the other LBM experiments

>>Reply:

Thank you for this valuable suggestion in terms of the flow and logic of the manuscript. We have now moved this paragraph forward to RL 204–211 according to your suggestion.

Minor comments/typos:

It would probably be good to be consistent with the labelling of subplots on figures. On some figures the panels a, b, c, d etc are in order going down each column, and on some they go left to right.

>>Reply:

Thank you very much for identifying this presentation inconsistency. We have now revised figures to be consistent in presentation layout.

Line 79: Typo in “eastern”

>>Reply:

Changed.

Line 85: Typo in “intensification”

>>Reply:

Changed.

Reviewer #3 (Remarks to the Author):

I appreciate the work that you have put in the paper, but I think that are substantial issues still left open.

>>Reply:

Thank you for your valuable comments that have further challenged our results and text, and compelled us to further improve the manuscript. Throughout our responses below, for convenience, the line numbers in the new revised manuscript are referred to as Revision Lines (RL).

1. The model description is still lacking, it is good to have now a description of the dissipation, but still is not described how the solution have been obtained, nor how the convergence has been monitored. Confirming results with a direct method would have been more convincing.

>>Reply:

In terms of obtaining the solution, according to Watanabe and Kimoto (2000, 2001), the LBM is based on the primitive equations. With a basic state \bar{X} , a steady response X follows an equation with the matrix form

$$L(\bar{X})X = F \quad (1)$$

where F indicates a forcing vector, and L indicates the linear dynamical operator that consists of the right-hand-sides of Equations 2–5. Note that X is what we show in Fig. 4 and related figures in the supplementary information.

The linearized equations for the perturbations are written as follows:

$$\frac{\partial \zeta'}{\partial t} = \frac{1}{a \cos \varphi} \frac{\partial A'_v}{\partial \lambda} - \frac{1}{a \cos \varphi} \frac{\partial}{\partial \varphi} (A'_u \cos \varphi) - \alpha \zeta' - K_v \left(\nabla^4 - \frac{2^2}{a^4} \right) \zeta' \quad (2)$$

$$\frac{\partial D'}{\partial t} = \frac{1}{a \cos \varphi} \frac{\partial A'_u}{\partial \lambda} + \frac{1}{a \cos \varphi} \frac{\partial}{\partial \varphi} (A'_v \cos \varphi) - \nabla_\sigma^2 (\Phi' + R[\bar{T}]\pi' + E') - \alpha D' - K_v \left(\nabla^4 - \frac{2^2}{a^4} \right) D' \quad (3)$$

$$\begin{aligned} \frac{\partial T'}{\partial t} = & -\frac{1}{a \cos \varphi} \frac{\partial}{\partial \lambda} (\bar{u}\bar{T}' + u'\bar{T}) - \frac{1}{a} \frac{\partial}{\partial \varphi} (\bar{v}\bar{T}' + v'\bar{T}) \cos \varphi + \bar{T}'\bar{D} + \bar{T}\bar{D}' - \bar{\sigma} \frac{\partial T'}{\partial \sigma} \\ & - \dot{\sigma}' \frac{\partial \bar{T}}{\partial \sigma} + \kappa T' \left(\frac{\partial \bar{\pi}}{\partial t} + \bar{V}_H \cdot \nabla_\sigma \bar{\pi} + \frac{\dot{\sigma}}{\sigma} \right) \\ & + \kappa \bar{T} \left(\frac{\partial \pi'}{\partial t} + V'_H \cdot \nabla_\sigma \bar{\pi} + \bar{V}_H \cdot \nabla_\sigma \pi' + \frac{\dot{\sigma}'}{\sigma} \right) - \alpha T' \\ & - K_h \left(\nabla^4 - \frac{2^4}{a^4} \right) T' \quad (4) \end{aligned}$$

$$\frac{\partial \pi'}{\partial t} = -\bar{V}_H \cdot \nabla_\sigma \pi' - V'_H \cdot \nabla_\sigma \bar{\pi} - \nabla_\sigma \cdot \bar{V}_H - \nabla_\sigma \cdot V'_H - \frac{\partial \bar{\sigma}}{\partial \sigma} - \frac{\partial \sigma'}{\partial \sigma} \quad (5)$$

where ζ , D , T , t , u , v , a , λ , φ , σ , and R represent the relative vorticity, divergence, temperature, time, meridional wind, zonal wind, earth radius, longitude, latitude, sigma level and universal gas constant, respectively. The Newtonian damping coefficient, α , in Equations 2–4 is set at 2 day^{-1} for the boundary layers $\sigma > 0.9$, 4 day^{-1} for levels $0.9 \gg \sigma > 0.8$, and zero elsewhere. K_v and K_h are the horizontal diffusion coefficients, which are set to be equal at $8 \times 10^{16} \text{ m}^4 \text{ s}^{-1}$. π is the logarithm of the surface pressure ($\ln P_S \equiv \pi$). $\kappa = R/c_p$, where c_p is the specific heat at constant pressure. V_H is the horizontal wind vector. E' is the perturbation of kinetic energy. The basic state and perturbations are denoted as $\bar{()}$ and $()'$, respectively.

$$A'_u = \zeta' \bar{v} + (\bar{\zeta} + f)v' - \bar{\sigma} \frac{\partial u'}{\partial \sigma} - \sigma' \frac{\partial \bar{u}}{\partial \sigma} - \frac{R\bar{T}}{a \cos \varphi} \frac{\partial \pi'}{\partial \lambda} - \frac{R\tilde{T}'}{a \cos \varphi} \frac{\partial \bar{\pi}}{\partial \lambda}, \quad (6)$$

$$A'_v = -\zeta' \bar{u} + (\bar{\zeta} + f)u' - \bar{\sigma} \frac{\partial v'}{\partial \sigma} - \sigma' \frac{\partial \bar{v}}{\partial \sigma} - \frac{R\bar{T}}{a} \frac{\partial \pi'}{\partial \varphi} - \frac{R\tilde{T}'}{a} \frac{\partial \bar{\pi}}{\partial \varphi}, \quad (7)$$

$$\Phi' = - \int_1^\sigma \frac{RT'_v}{\sigma} d\sigma, \quad (8)$$

$$\begin{aligned} \dot{\sigma}' = & \sigma \int_0^1 \bar{V}_H \cdot \nabla_\sigma \pi' d\sigma + \sigma \int_0^1 V'_H \cdot \nabla_\sigma \bar{\pi} d\sigma - \int_0^\sigma \bar{V}_H \cdot \nabla_\sigma \pi' d\sigma - \int_0^\sigma V'_H \cdot \nabla_\sigma \bar{\pi} d\sigma \\ & + \sigma \int_0^1 D' d\sigma - \int_0^\sigma D' d\sigma \quad (9) \end{aligned}$$

f in Equations 6–7 is the Coriolis parameter (a.k.a. the planetary vorticity).

In terms of the convergence of simulations, previous studies (e.g., An et al., 2022; Lu and Lin, 2009; Watanabe and Jin, 2003) have shown that integrations for 15–20 days are enough to produce a steady response. In this paper, we integrated the model for 20 days, with results averaged over the 16–18th days taken as the steady response to the prescribed heating. Detailed evolutions of each experiment were also shown in Extended Data Figs. 7–11, which further confirmed the convergence of our simulations after day 15.

Following the above descriptions, we have now added relevant explanations in the revised manuscript (RL 300–322).

2. The structure of the heating is still puzzling. It is also unclear how the vertical structure of the heating has been derived. Even considering the imposed heating (red line in Fig. 4) it is concentrated at 200mb, implying really high level anomalous convection that is not typical of the Mediterranean area. In any case, the balances at this latitude require that heating is compensated by meridional motion, not vertical motion, negating a Gill-like response that would generate a high level vorticity source. So it is likely that the response that you see is generated by a quasi-resonant response of the linear model, due to ill-conditioning of the linear system. Evidence of this can be seen in the almost identical response in panels (b) and (e) except for the sign that of course is a consequence of the forcing having two different signs. By the way, it is also strange that, given the similarity in the 500mb Geopotential the SLP is different.

>>Reply:

Thank you for this insightful comment.

- (1) The vertical structure of the heating was calculated as a residual based on the temperature tendency equation, which has been detailed in the manuscript (RL 289–298).
- (2) The heating profile, presented in Fig. 4, was the **anomalous** heating. Here we show the total heating composited for the warm blobs, the climatological heating, and the anomalous heating in Fig. R12. The total heating (yellow line) reaches the maximum value near 700-600 hPa, which is the shallow heating, in good agreement with your comment. However, we imposed an anomalous heating (blue line), with a maximum at 300 hPa, in the LBM based on the linearized primitive equations. In fact, the maximum **anomalous** heating level was located at 500 hPa if we used daily data to calculate the diabatic heating term and then averaged within November (not shown). In addition, the Mediterranean region can have convection activity in November with heating located at higher levels (Tuduri and Ramis, 1997; Khodayar et al., 2018).

Fig. R12 Total heating (yellow) and anomalous (blue) heating composited for warm blobs in November and corresponding climatological heating (green) averaged over the Mediterranean region.

- (3) We think the Gill theory cannot be appropriately applied in the mid-latitude regions. The LBM replicates the anomalous ridge well in the Northeast Pacific when imposed forcing is applied over the Mediterranean and North Atlantic regions. Nevertheless, we agree that there are some limitations of the LBM. Thus, we further performed CAM5 simulations based on more complicated and realistic physical processes. In addition, the negative SLP anomalies in North Atlantic experiment (Fig. 4f) were well in accordance with the anomalous trough to the west (Fig. 4e) due to baroclinic response.

The nonlinear experiments in Fig. 5 do not help much. First of all, the experiments are not described (which climatology is subtracted? how they have been initialized?), then they show that basically the perturbations tend to generate instabilities in the most sensitive areas of the jets, over the Pacific/North American sector and over the Atlantic. Also in this case, it is hard to judge that the Med and Atl response are significantly different, for instance, from putting perturbations in any other area.

>>Reply:

The CTRL experiment was designed using monthly climatological SST over 1981–2010 globally. Sensitivity experiments were forced by anomalous SST superimposed onto climatological SST in the specific focused region. Then, we show anomalies between the sensitivity experiments and CTRL experiment in Fig. 5 and Extended Data Fig. 10.

Each experiment was initialized from 1 January 1979 with prescribed SST. The model

will read its own file with initial conditions on 1 January 1979. Each experiment was integrated for 25 years, with the last 20 years averaged for our analyses. The first 5 years were not used for these analyses, because this represents the model spin-up period, which also minimized the effects of the initial condition.

To further demonstrate the combined role of the Mediterranean and North Atlantic regions in generating circulation anomalies downstream, we conducted an additional experiment with SST forcing over the tropical North Atlantic (EXP_TNA) (Fig. R13). Although perturbations tend to generate instabilities in most sensitive areas of the jets, we can see clear differences between this new experiment and EXP_NAtl experiment in the manuscript (Figs. R13 and 5d–f).

Accordingly, we modified the related model set-up descriptions in RL 324–342.

Fig. R13 Difference of geopotential height response (shading, in gpm) at 500 hPa between tropical North Atlantic experiment (EXP_TNA) and NATL experiment (Exp_NAtl).

In summary, I think that the point of the remote influence of the Med on the North Pacific has not been sufficiently demonstrated.

>>Reply:

Thank you very much again for your valuable comments. We hope that our results and responses now provide a more compelling demonstration of the mechanisms.

References:

- An, X., Sheng, L., Li, C., Chen, W., Tang, Y., & Huangfu, J. Effect of rainfall-induced diabatic heating over southern China on the formation of wintertime haze on the North China Plain. *Atmos. Chem. Phys.* 22, 725–738 (2022).
- Lu, R., & Lin, Z. Role of subtropical precipitation anomalies in maintaining the summertime meridional teleconnection over the western North Pacific and East Asia. *J. Clim.* 22, 2058–2072 (2009).

- Watanabe, M., & Jin, F. F. A moist linear baroclinic model: Coupled dynamical–convective response to El Niño. *J. Clim.* 16, 1121–1139 (2003).
- Watanabe, M., & Kimoto, M. Atmosphere-ocean thermal coupling in the North Atlantic: A positive feedback. *Quart. J. Roy. Meteor. Soc.* 126, 3343–3369 (2000).
- Watanabe, M., & Kimoto, M. Corrigendum. *Quart. J. Roy. Meteor. Soc.* 127, 733–734 (2001).
- Tudurí, E., & Ramis, C. The Environments of Significant Convective Events in the Western Mediterranean. *Wea. Forecasting*, 12, 294–306 (1997).
- Khodayar, S., Kalthoff, N., & Kottmeier, C. Atmospheric conditions associated with heavy precipitation events in comparison to seasonal means in the western mediterranean region. *Clim. Dyn.* 51, 951–967 (2018).

REVIEWER COMMENTS

Reviewer #1 (Remarks to the Author):

I thank the authors' efforts answering my questions. For comment 2, could the authors check the composite atmospheric circulation patterns lagged each blob events? If the circulation patterns are consistent, then this mechanism should work. I do not have further comments.

Reviewer #2 (Remarks to the Author):

I am now satisfied that the authors have addressed my concerns.

Reviewer #3 (Remarks to the Author):

I understand the hard work the authors did to try to improve the paper, but I feel that still there are issues:

1. The calculation of the heating is still not clear as they do not say on which data set the residual were calculated.
2. The issue of compensation of heating in the mid latitudes is standard QG dynamics (Hoskins and Karoly, 1981)
3. Allow me to differ on the convergence rate. Allow me to differ on the convergence interpretation. The near field response is stable, but the far field response is not, and that, of course, is what matters in the paper. I think that it would be much better to obtain a direct (i.e. matrix inversion) linear solution, eliminating issues of convergence.
4. You have to double the amplitude of the heating in the Med and Atlantic experiments. It would be fine in the attempt to understand the signature of possible effects, but in terms of assessing the role of this heating in the real atmosphere it sort of show that the impact is minor.

In conclusion, the paper is vastly improved, but I think that the impact of the Mediterranean region as a source of anomalous heating for the North Pacific has not been conclusively demonstrated. It can still be published, removing the statements and the conclusions in this direction.

REVIEWER COMMENTS

Reviewer #1 (Remarks to the Author):

I thank the authors' efforts answering my questions. For comment 2, could the authors check the composite atmospheric circulation patterns lagged each blob events? If the circulation patterns are consistent, then this mechanism should work. I do not have further comments.

>>Reply:

We very much appreciate the reviewer's positive comments. We further checked the composite atmospheric circulation patterns lagged warm blob events. An anomalous trough occurred over the Alaska region, which was statistically significant.

Fig. R1 As in Fig. 1, but for lagging one month of the peak.

Reviewer #2 (Remarks to the Author):

I am now satisfied that the authors have addressed my concerns.

>>Reply:

Thanks very much.

Reviewer #3 (Remarks to the Author):

I understand the hard work the authors did to try to improve the paper, but I feel that still there are issues:

>>Reply:

We greatly appreciate the reviewer's constructive comments that have helped us further improve the manuscript. Throughout our responses below, for convenience, the line numbers in the new revised manuscript are referred to as Revision Lines (RL).

1. The calculation of the heating is still not clear as they do not say on which data set the residual were calculated.

>>Reply:

The calculation of the heating is based on data from the ERA5 dataset. We added this information in RL 249 and 292.

2. The issue of compensation of heating in the mid latitudes is standard QG dynamics (Hoskins and Karoly, 1981).

>>Reply:

Thank you for reminding us about this paper, which helps understand the QG-related compensation of heating in the mid latitudes. We have now cited this paper in RL 119 and 194.

3. Allow me to differ on the convergence rate. Allow me to differ on the convergence interpretation. The near field response is stable, but the far field response is not, and that, of course, is what matters in the paper. I think that it would be much better to obtain a direct (i.e. matrix inversion) linear solution, eliminating issues of convergence.

>>Reply:

Thank you for this suggestion. However, we stress that, according to previous studies and our LBM daily outputs (Extended Data Figs. 7 and 9), the response over the Northeast Pacific region is generally stable after about 16 days. With respect, we do not see the value or necessity to take this further given the stability of our analysis already undertaken and outlined.

4. You have to double the amplitude of the heating in the Med and Atlantic experiments. It would be fine in the attempt to understand the signature of possible effects, but in terms of assessing the role of this heating in the real atmosphere it sort of show that the impact is minor.

>>Reply:

We have doubled the amplitude of the heating to better highlight the signature of the effects from the Mediterranean and North Atlantic. However, the response over the Northeast Pacific Ocean (NEP) is very similar to that produced without doubling the heating, as per our results in the previous version of our manuscript.

In addition, we show that the contribution from the tropics may not be large enough in November, compared to that in December to February. The geopotential height

response over the NEP forced by the tropics is negligible or comparable to the forcing from the mid-latitudes (Figs. R1–R3).

Fig. R1 As in Extended Data Fig. 7, but imposing anomalous cooling over the Maritime Continent.

Fig. R2 As in Extended Data Fig. 7, but imposing anomalous heating over the tropical Indian Ocean.

Fig. R3 As in Extended Data Fig. 7, but imposing anomalous heating over the tropical western Pacific.

In conclusion, the paper is vastly improved, but I think that the impact of the Mediterranean region as a source of anomalous heating for the North Pacific has not been conclusively demonstrated. It can still be published, removing the statements and the conclusions in this direction.

>>Reply:

Thank you again for your suggestions and recommendations to improve the paper. We believe we have provided a compelling demonstration of important contributions from anomalous heating in the Mediterranean/NATL regions to the ridge over the NEP. However, we respectfully acknowledge and appreciate the reviewer's concerns, and consequently we have now toned down the strength of our statements regarding the role of the Mediterranean.